# Incidence of common infectious diseases in Japan during the COVID-19 pandemic

**Kenji Hibiya**[1,2]�*, **Hiroyoshi Iwata**[3]�, **Takeshi Kinjo**[1‡], **Akira Shinzato**[1‡],
**Masao Tateyama**[1‡], **Shinichiro Ueda**[3‡], **Jiro Fujita**[1]�

**1** Department of Infectious, Respiratory and Digestive Medicine, Graduate School of Medicine, University of
the Ryukyus, Nishihara-cho, Okinawa, Japan, **2** Department of Diagnostic Pathology, University of the
Ryukyus Hospital, Nishihara-cho, Okinawa, Japan, **3** Clinical Pharmacology & Therapeutics, University of
The Ryukyus School of Medicine, Nishihara-cho, Okinawa, Japan

he These authors contributed equally to this work.
‡ TK, AS, MT and SU also contributed equally to this work.
* h2139517@med.u-ryukyu.ac.jp

University College Station, UNITED STATES

**Data Availability Statement:** All relevant data are
within the manuscript and its Supporting
Information files.

**Funding:** The authors received no specific funding
for this work.

## Abstract

Recent reports indicate that respiratory infectious diseases were suppressed during the
novel coronavirus disease-2019 (COVID-19) pandemic. COVID-19 led to behavioral
changes aimed to control droplet transmission or contact transmission. In this study, we
examined the incidence of common infectious diseases in Japan during the COVID-19 pan-
demic. COVID-19 data were extracted from the national data based on the National Epide-
miological Surveillance of Infectious Diseases (NESID). Common infectious diseases were
selected from notifiable infectious diseases under the NESID. The epidemic activity of the
diseases during 2015–2020 was evaluated based on the Infectious Disease Weekly
Reports published by the National Institute of Infectious Diseases. Each disease was then
categorized according to the route of transmission. Many Japanese people had adopted
hygienic activities, such as wearing masks and hand washing, even before the COVID-19
pandemic. We examined the correlation between the time-series of disease counts of com-
mon infectious diseases and COVID-19 over time using cross-correlation analysis. The
weekly number of cases of measles, rotavirus, and several infections transmitted by droplet
spread, was negatively correlated with the weekly number of cases of COVID-19 for up to
20 weeks in the past. According to the difference-in-differences analysis, the activity of influ-
enza and rubella was significantly lower starting from the second week in 2020 than that in
2015–2019. Only legionellosis was more frequent throughout the year than in 2015–2019.
Lower activity was also observed in some contact transmitted, airborne-transmitted, and
fecal-oral transmitted diseases. However, carbapenem-resistant *Enterobacteriaceae*, exan-
thema subitum, showed the same trend as that over the previous 5 years. In conclusion, our
study shows that public health interventions for the COVID-19 pandemic may have effec-
tively prevented the transmission of most droplet-transmitted diseases and those transmit-
ted through other routes.

**Competing interests:** The authors declare no competing financial interests.

## Introduction

The novel coronavirus, named severe acute respiratory syndrome coronavirus 2 (SARS-CoV-2), emerged in Wuhan, Hubei province, China, in late 2019 and has since spread worldwide through the transnational movement of people [1]. In Japan, the first case of SARS-CoV-2 infection, known as coronavirus disease 2019 (COVID-19), was confirmed on January 16, 2020. The infection spread among tourists and returnees from China and their close contacts [2]. In response to the COVID-19 pandemic, behavioral modification encompassing wearing masks, handwashing, and avoiding crowded spaces was encouraged [2]. Considering that SARS-CoV-2 is primarily transmitted through respiratory droplets and contact [3], these behaviors might decrease the spread of COVID-19 and other common infectious diseases [4]. Notably, the activity of seasonal influenza in 2020 was lower than that in 2019 in Japan [5, 6]. However, the activities and trends of other infectious diseases during the COVID-19 pandemic have not been evaluated. Therefore, the present study examined the activities of common infectious diseases in Japan during the COVID-19 pandemic based on nationwide surveillance by the Ministry of Health, Labour, and Welfare.

## Material and methods

### Datasets about COVID-19

The COVID-19 pandemic from January 16, 2020, to December 31, 2020 was statistically analyzed using datasets from the National Epidemiological Surveillance of Infectious Diseases (NESID) under the Infectious Diseases Control Law. We obtained the open data about COVID-19 from January 16, 2020 to December 31, 2020 from the Ministry of Health, Labour and Welfare [7]. Data on the daily number of new positive cases who have taken polymerase chain reaction for SARS-CoV-2 or antigen testing for SARS-CoV-2 were used in this study. These domestic cases do not include cases of airport quarantine.

### Datasets of common infectious diseases

Common infectious diseases were selected from the nationally notifiable diseases, according to the following: i) we excluded diseases with <400 cases of infection per year. Since there are 365 days in a year, we set the number to >400, considering more than one case per day. ii) We excluded fulminant and invasive infectious diseases, such as invasive pneumonia disease, invasive meningococcal disease, and severe invasive streptococcal disease. The total number of invasive infections is small. However, invasive infections indicate the severity of the disease and do not necessarily reflect the frequency or route of infection. Therefore, we deleted those from the analysis. iii) We excluded "infectious gastroenteritis," a syndrome induced by various causes, such as bacteria, viruses, and parasites. Difficulties arise when classifying it via the transmission route. Thus, it was excluded. iv) In addition, we excluded monthly reports of infections, such as gonococcal infections or multi-drug-resistant *Pseudomonas aeruginosa* infection. Common infectious diseases were divided into two groups: i) diseases from the sentinel surveillance system and ii) diseases from the passive surveillance system (Table 1). Sentinel surveillance systems involve clinics or hospitals or public health centers, local infectious disease surveillance centers (local IDSCs), and the national infectious disease surveillance center (national IDSC). The public health center gathers a total number of patients during 1 week with target diseases diagnosed at each medical facility with influenza, pediatric, ophthalmic, and designated sentinel sites. Local IDSCs gather the data from public health centers in a prefecture. The national IDSC then gathers the data from the local IDSC. The weekly number of cases includes number of cases diagnosed at each facility per week divided by the number of

**Table 1. Classification of common infectious diseases by the Japanese surveillance system.**

| Category | Source | Name of diseases |
|---|---|---|
| **Sentinel surveillance** | Influenza sentinel sites (approximately 5,000) | Influenza (excluding avian influenza) |
| | Pediatric sentinel sites (approximately 3,000) | Respiratory syncytial virus |
| | | Pharyngoconjunctival fever |
| | | Group A Streptococcal pharyngitis |
| | | Chicken pox (varicella) |
| | | Hand, foot, and mouth disease |
| | | Erythema infectiosum |
| | | Exanthema subitum |
| | | Herpangina |
| | | Mumps |
| | Ophthalmology sentinel sites (approximately 700) | Epidemic keratoconjunctivitis |
| | Designated sentinel sites (approximately 500) | *Mycoplasma pneumoniae* pneumonia |
| | | Infectious gastroenteritis (rotavirus) |
| **Passive surveillance** | All medical care facilities (179,475*) | Tuberculosis |
| | | Enterohemorrhagic *Escherichia coli* infection |
| | | Hepatitis A |
| | | Hepatitis E |
| | | Scrub typhus |
| | | Legionellosis |
| | | Amoebiasis |
| | | Carbapenem-resistant *Enterobacteriaceae* |
| | | Acquired immunodeficiency syndrome |
| | | Syphilis |
| | | Pertussis (whooping cough) |
| | | Rubella |
| | | Measles |

*Numbers at the end of September 2020.

facilities with sentinel sites. This number of sentinel sites reflects Japan's overall trends regarding infectious disease epidemics. Sentinel sites are set under the jurisdiction of the public health center such that the nationwide morbidity rate can be estimated with a standard error rate ≤5%. In contrast, passive surveillance systems gather data from all domestic medical care facilities in Japan.

The actual reported number of infectious disease cases was collected from the Infectious Diseases Weekly Reports in accordance with the NESID, which is conducted by the National Institute of Infectious Diseases [8]. The data was added in S1 Table. Though the actual reported numbers were reported per week, this "week" indicates an epidemiological week. For example, the first week in 2020 is from December 30 to January 5, and the first week in 2019 is from December 31 to January 6. This is referred to as the "weeks ending log" prescribed by the National Institute of Infectious Diseases (see https://www.niid.go.jp/niid/en/calendar-e.html).

## Categorization of infectious disease based on routes of transmission

Common infectious diseases were categorized according to the transmission routes (Table 2) based on the guidelines of the Centers for Disease Control and Prevention [9]. We emphasized transmission routes that are important for infection prevention in the community setting for

**Table 2. The incidence of common infectious diseases in 2020 and 2019 based on transmission routes.**

| Infection transmission routes | Name of diseases | 2019 | 2020 | Ratio (2020/2019) |
|---|---|---|---|---|
| | | No. per sentinel / *Total number**[*] | No. per sentinel / *Total number**[*] | |
| **Droplet** | Influenza | 379.73 | 114.27 | 0.30 |
| | Respiratory syncytial virus | 44.38 | 5.74 | 0.13 |
| | Pharyngoconjunctival fever | 23.91 | 11.14 | 0.47 |
| | Group A Streptococcal pharyngitis | 112.51 | 63.52 | 0.56 |
| | Hand, foot, and mouth disease | 127.54 | 5.83 | 0.05 |
| | Erythema infectiosum | 34.29 | 5.79 | 0.17 |
| | Herpangina | 30.76 | 8.02 | 0.26 |
| | Mumps | 4.80 | 2.56 | 0.53 |
| | *Mycoplasma pneumoniae* pneumonia | 12.67 | 7.36 | 0.58 |
| | Legionellosis | *2 316* | *2 031* | 0.88 |
| | Pertussis (whooping cough) | *16 845* | *2 932* | 0.17 |
| | Rubella | *2 298* | *100* | 0.04 |
| **Contact** | Exanthema subitum | 20.44 | 20.79 | 1.02 |
| | Epidemic keratoconjunctivitis | 33.25 | 13.08 | 0.39 |
| | Carbapenem-resistant *Enterobacteriaceae* | *2 333* | *1 922* | 0.82 |
| **Airborne** | Chicken pox (varicella) | 18.00 | 10.08 | 0.56 |
| | Tuberculosis | *21 672* | *17 108* | 0.79 |
| | Measles | *744* | *13* | 0.02 |
| **Fecal-oral** | Infectious gastroenteritis (rotavirus) | 9.82 | 0.52 | 0.05 |
| | Enterohemorrhagic *Escherichia coli* infection | *3 744* | *3 064* | 0.82 |
| | Hepatitis A | *425* | 119 | 0.28 |
| | Hepatitis E | *490* | *450* | 0.92 |
| **Vector-borne** | Scrub typhus | *404* | *511* | 1.26 |
| **Sexual** | Amoebiasis | *853* | *610* | 0.72 |
| | Acquired immunodeficiency syndrome | *1 231* | *1 075* | 0.87 |
| | Syphilis | *6 642* | *5 784* | 0.87 |

[*]The vertical bar represents the number of patients referred to one medical facility per week per selected time point, and italic figures show the total number by 100% survey.

pathogens with multiple human transmission routes [10]. For example, hepatitis A has three human transmission routes: i) fecal-oral, ii) contact, and iii) sexual [10]. However, we categorized hepatitis A as a fecal-orally transmitted disease. Although rotavirus has an aspect of contact transmission [10], we also categorized it as a fecal-orally transmitted disease. In addition, although amoebiasis has an aspect of fecal-oral transmission, we categorized it as a sexually transmitted disease (STD).

## Total number of outpatient cases in the medical clinics

The total number of outpatient cases in the domestic medical clinics was obtained from the Ministry of Health, Labor and Welfare's estimated medical expenses database [11]. "Medical clinic" means the places where doctors provide medical practice and do not have hospitalization facilities for patients or have hospitalization facilities for <19 patients. The number of cases referred to is the number of medical fee statements (receipts), and each medical institution prepares one statement for one patient every month. The data was added as S2 Table.

## Statistical analyses

For the epidemic curves of "COVID-19" and "common infectious diseases", a moving average with 2 points was calculated. The cross-correlation functions (CCFs) were used to understand whether there was a time-lagged correlation between common infectious disease count and COVID-19 incidence. CCF is a function that expresses the similarity between two-time series and gives information about how similar and displaced one-time series is to the other. CCF takes a value in the range of -1 (negative correlation) to 1 (positive correlation). If the correlation value exceeds the confidence level, then the two series are correlated. The cross-correlation between the two variables is statistically significant at approximately 5% level of significance. To compare the time series change in case numbers of common infectious diseases in 2020 with that in the previous 5 years (2015–2019), a difference-in-differences linear regression was applied for infectious diseases transmitted by droplets that are most susceptible to preventative behavioral changes. The model included a categorical variable for each week, a categorical variable for the 2020 season (versus the 2015–2019 seasons) and the interaction variables for each week and the 2020 season, following the method described by Sakamoto et al. (2020) [5]. We made two assumptions for the difference-in-difference linear regression in our preliminary experiments. First, the parallel trend assumption was valid for both incidences because the current incidents and incidences of the previous 5 years of common infections were parallel. The common shock assumption was also valid, as it showed a similar change when an event (the epidemic of COVID-19) occurred, indicated the appropriateness of this study design. For the statistical analysis, Light Stone® STATA® ver.15 was used.

## Results

### Outbreak of COVID-19 in Japan

In Japan, COVID-19 showed three waves of outbreak in 2020. The first peak of the COVID-19 outbreak was marked on April 11 (Fig 1). In response to this outbreak, the government declared a state of emergency on April 16, 2020 [12]. The government restricted various human behaviors, including the regulation of transnational/transborder traveling and mass gathering, temporary closure of all Japanese elementary, junior high, and senior high schools, and implementation of a remote working model [13, 14]. The incidence of COVID-19 was steady at below 100 cases from mid-May 2020 to mid-June 2020 (Fig 1). However, the disease began to spread again at the end of June 2020. Despite the rising cases, the "Go to travel" operation to enhance domestic trips was initiated on July 22, 2020 [15]. On August 7, 2020, the pandemic curve showed the second peak of the COVID-19 outbreak attributed to travelers and returnees from Europe or the United States [16] (Fig 1). Although the pandemic steadied at around 500 new cases per day, the cases increased again from November 2020 to form the third wave (Fig 1). On December 31, 2020, Japan experienced a resurgence of COVID-19.

### The trend of common infectious diseases in Japan, 2020

The epidemic curves of each common infectious disease compared with those of COVID-19 are shown in Fig 2A and 2B. Influenza, pharyngoconjunctival fever, group A streptococcal pharyngitis, chicken pox, erythema infectiosum, epidemic keratoconjunctivitis, *Mycoplasma pneumoniae* pneumonia, and pertussis showed an epidemic before the COVID-19 outbreak (Fig 2A and 2B). However, respiratory syncytial virus (RSV) showed only minor activity before the COVID-19 outbreak (Fig 2A).

Hand-foot-and-mouth disease (HFMD), infectious gastroenteritis, rubella, and measles did not show an obvious epidemic trend amid the COVID-19 outbreak, and the number of patients

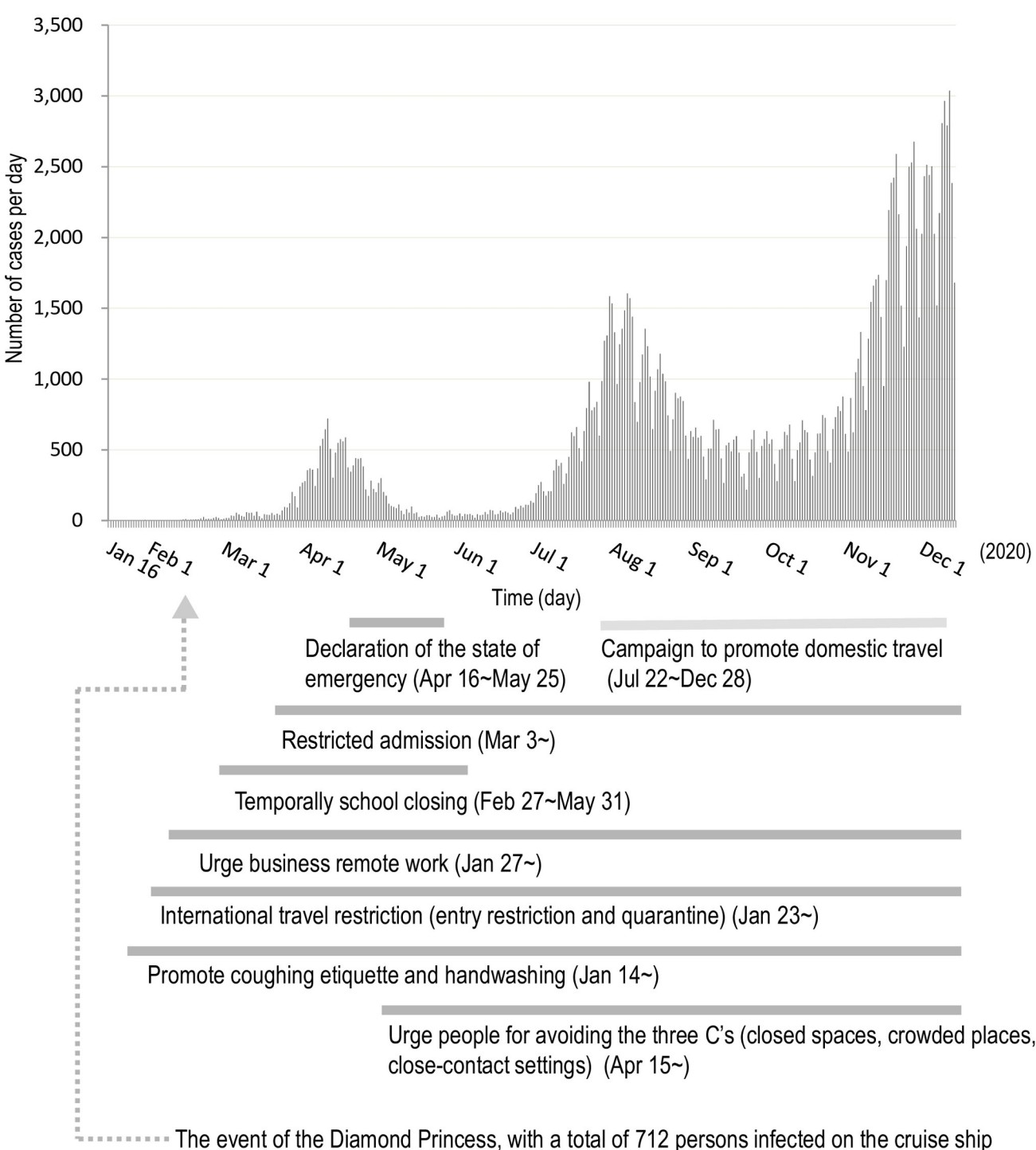

**Fig 1. Daily new confirmed COVID-19 cases in Japan.** The figure is based on the domestic infection status officially released by the Ministry of Health, Labour and Welfare in Japan [7].

with HFMD and infectious gastroenteritis was maintained under 0.1 per sentinel site throughout the year (Fig 2A and 2B). Herpangina showed a mild epidemic in the summer season, corresponding to the second wave of COVID-19 (Fig 2A). Exanthema subitum showed an epidemic trend in the summer season, similar to the previous year, regardless of the COVID-19 outbreak

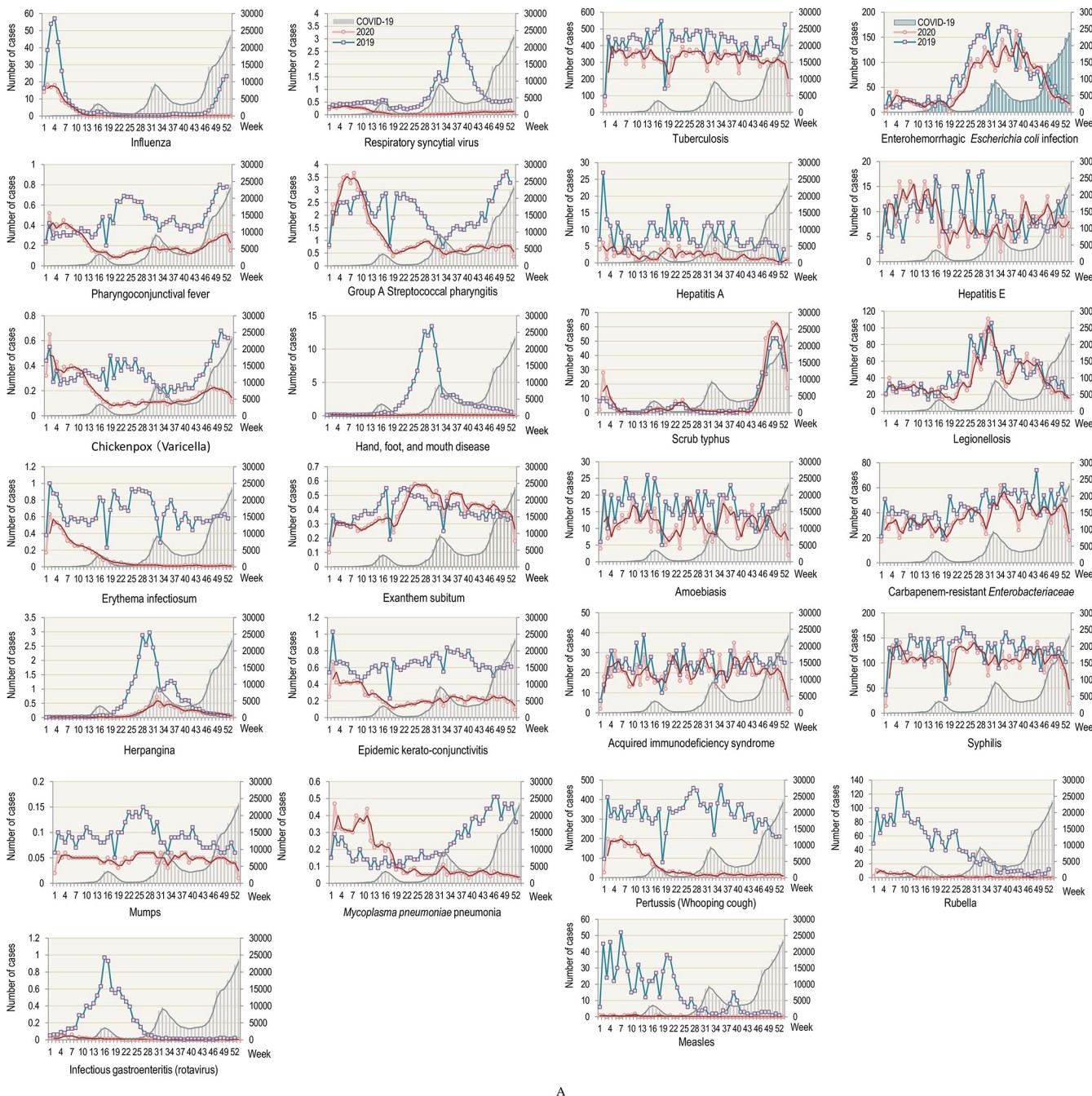

**Fig 2. Current epidemic curves of common infectious diseases in 2020 compared with those in 2019.** (a) Common infectious diseases under the national sentinel surveillance. (b) Common infectious diseases under the national notifiable disease surveillance. Each left vertical axis represents the number of patients referred to one medical facility per week per selected time point (a) or the total number of cases per week reported (b). Each right vertical axis and gray bars () indicate the total number of newly confirmed COVID-19 cases per week reported in Japan. The blue lines () indicate the 2019 epidemic curves. The scarlet thin lines () indicate the 2020 epidemic curve. The moving average lines were applied to the column graph of COVID-19 incidence (gray tick line) and to the incidence curve of 2020 (scarlet tick line) of each common infectious disease. The "week" along the x-axis indicates the epidemiological week. Data on COVID-19 were obtained from surveillance data by the National Institute of Infectious Diseases in Japan [8]. Data on common infectious diseases were obtained from the Infectious Diseases Weekly Reports from the National Epidemiological Surveillance of Infectious Diseases by the National Institute of Infectious Diseases in Japan [8].

(Fig 2A). Scrub typhus showed an epidemic trend in the winter season as in the previous year, corresponding to the third wave of COVID-19 (Fig 2B). Enterohemorrhagic *Escherichia coli* (EHEC) and legionellosis showed an epidemic trend as in the previous year, corresponding to the second wave of COVID-19 (Fig 2B). The trends of tuberculosis, hepatitis A, hepatitis E, amoebiasis, carbapenem-resistant *Enterobacteriaceae* (CRE), acquired immune deficiency syndrome (AIDS), and syphilis were not consistent with the COVID-19 outbreak (Fig 2B).

The epidemic curves of common infectious diseases in 2020 were compared with those in 2019 (Fig 2A and 2B). Some diseases such as influenza, RSV, pharyngoconjunctival fever, group A Streptococcus pharyngitis, chicken pox, HFMD, erythema infectiosum, herpangina, epidemic keratoconjunctivitis, mumps, *Mycoplasma pneumoniae* pneumonia, infectious gastroenteritis, hepatitis A, pertussis, rubella, and measles were suppressed compared with the previous year. Others such as exanthema subitum, tuberculosis, EHEC, hepatitis E, scrub typhus, legionellosis, amoebiasis, CRE, AIDS, and syphilis had similar epidemic trends as the previous year.

The former group which was suppressed compared with the previous year can be further divided into two sub-groups: i) diseases with epidemic trends before the COVID-19 outbreak and ii) diseases with no epidemic trends in 2020. Influenza, pharyngoconjunctival fever, group A streptococcal pharyngitis, chicken pox, erythema infectiosum, epidemic keratoconjunctivitis, *Mycoplasma pneumoniae* pneumonia, and pertussis showed an epidemic trend before the COVID-19 outbreak, and the ratio of cases in 2020 to those in 2019 ranged from 0.17 to 0.56 (Table 2). For HFMD, infectious gastroenteritis, rubella, and measles that did not show an obvious epidemic trend, the ratio of cases in 2020 to those in 2019 was ≤0.05 (Table 2). The number of patients with mumps in 2020 was lower than in 2019 and showed steady activity throughout 2020 (Fig 2A). The number of patients with hepatitis A in 2020 was lower than in 2019, although there was little activity until week 33.

In the latter group which had similar epidemic trends as the previous year, exanthema subitum and scrub typhus were more active in 2020 than in 2019 (Fig 2A). The ratio of the number of patients in 2020 to those in 2019 was over 1.00 (Table 2). The number of patients with tuberculosis, EHEC, hepatitis E, legionellosis, amoebiasis, CRE, AIDS, and syphilis in 2020 decreased slightly compared with that in 2019 (Fig 2B). The ratio of patients in 2020 to those in 2019 ranged from 0.72 to 0.92 (Table 2).

Additionally, we examined the CCF between the two-time series of COVID-19 and common infectious disease counts (S1 Fig). The incidence of scrub typhus peaked significantly 1 week earlier than the third peak of COVID-19 incidence (cross-correlation values = 0.87). Herpangina showed a significant peak 15 weeks earlier than the second peak time of COVID-19 incidence (cross-correlation values = 0.65). The strongest negative correlation (cross-correlation values = -0.40) was obtained at lag 1 week for mumps. Influenza, group A Streptococcal pharyngitis, erythema infectiosum, epidemic keratoconjunctivitis, *Mycoplasma pneumoniae* pneumonia, infectious gastroenteritis (rotavirus), pertussis, rubella, measles showed a negative correlation with COVID-19 in the lag from minus 20 weeks to 0 weeks. Sexually transmitted diseases, including amoebiasis, AIDS, and syphilis, reached their lowest peaks 1 to 2 weeks later than the peak in COVID-19 incidence. A closer look at the epidemic curve showed phenomena with a slight deviation from each peak of COVID-19.

## Relationship between epidemic curves and transmission routes

Each common infectious disease was categorized based on the main transmission route (Fig 3, Table 2).

In the category of droplet transmission, all diseases except legionellosis had a lower epidemic curve in 2020 than in 2015–2019 (Fig 3A).

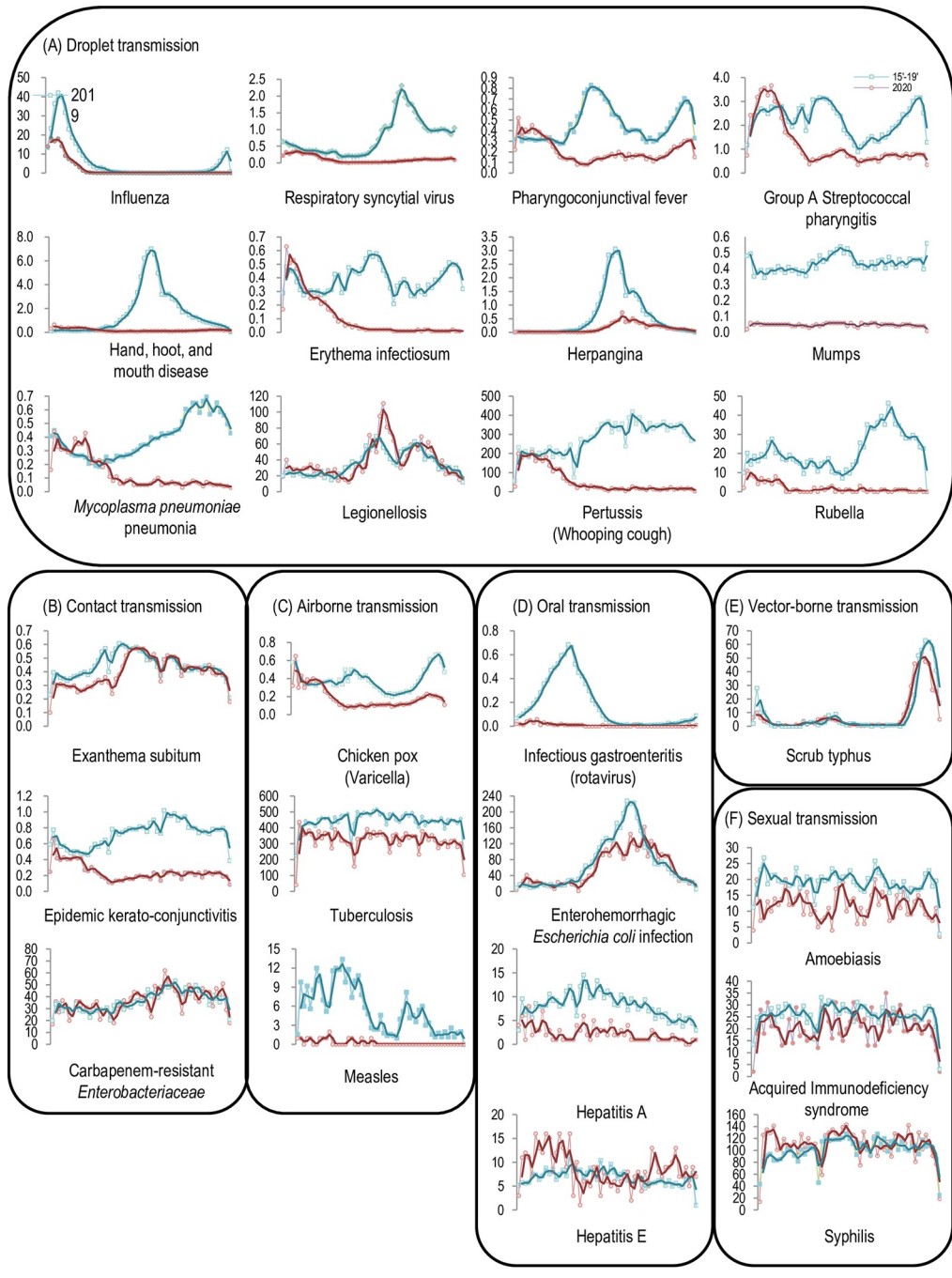

**Fig 3. Epidemic curves of common infectious diseases classified according to the transmission routes.** The blue thin lines () indicate the epidemic curves during 2015–2019. The scarlet thin lines () indicate the 2020 epidemic curves. The moving average lines were applied to the incidence curve of 2020 (scarlet tick line) and 2015–2019 (blue tick line) of each common infectious disease. Data were obtained from the Infectious Diseases Weekly Reports from the National Epidemiological Surveillance of Infectious Diseases by the National Institute of Infectious Diseases in Japan [8].

In the category of contact transmission, epidemic keratoconjunctivitis showed lower activity in 2020 than in 2015–2019, although exanthema subitum and CRE had comparable epidemic curves in 2015–2019 and 2020 (Fig 3B).

In the category of airborne transmission, varicella and measles showed lower activity in 2020 than in 2015–2019 (Fig 3C). The epidemic curves of tuberculosis in 2015–2019 and 2020 were similar but comparatively lower (Fig 3C).

In the category of fecal-oral transmission, each disease showed different trends (Fig 3D). Infectious gastroenteritis did not show any activity in 2020. EHEC showed an epidemic curve resembling that of 2015–2019. Hepatitis A showed lower activity in 2020 compared with 2015–2019. Hepatitis E was less active in the summer of 2020 but was more active during the rest of the year compared with 2015–2019.

In the category of vector-borne transmission, scrub typhus in 2020 showed an epidemic curve resembling that of 2015–2019 (Fig 3E).

In the category of sexual transmission, epidemic curves of amoebiasis, AIDS, and syphilis in 2020 resembled those of 2015–2019 (Fig 3F).

According to the difference-in-differences analysis, the activity of influenza was significantly lower since the second week in 2020 than during 2015–2019 (Fig 4). Similarly, respiratory syncytial virus was lower after 27 weeks, Group A streptococcal pharyngitis was lower after 10 weeks. Hand, foot, and mouth disease was lower after 5 weeks, erythema infectiosum was lower after 13 weeks. Herpangina was lower after 11 weeks, mumps was lower during 27 to 30 weeks, *Mycoplasma pneumoniae* pneumonia was lower after 25 weeks, pertussis was lower after 15 weeks, and rubella was lower after 31 weeks (Fig 4). However, legionellosis was more frequent throughout the year than during 2015–2019 (Fig 4).

## Relationship between the COVID-19 epidemic and total number of outpatient cases

The total number of outpatients is shown year-on-year in the same month (Fig 5, S2 Table). At the same time, the epidemic curves of COVID-19 were superimposed. The number of outpatients in any clinical departments decreased in May 2020 and September 2020. The decrement was the greatest in pediatrics. The transition was similar to the COVID-19 epidemic curve.

## Discussion

### Behavior modification to control COVID-19 in Japan

The preventative behaviors of wearing masks and hand hygiene prevailed in the early stages of the COVID-19 pandemic in Japan. For example, face masks and hand sanitizers were sold out in weeks 3–4 of January 2020 [17]. In January 2020, sales of masks increased five-fold compared with the same month in the previous year, and that of hand sanitizers increased six-fold compared with the same month in the previous year [18]. Muto et al. (2020) reported frequent hand washing by 86% of Japanese participants (n = 11,342) during the early phase of the pandemic [2]. Additionally, in March 2020, the Prime Minister advised the public to avoid the three Cs (closed spaces, crowded places, and close-contact settings) to avert the clustering of COVID-19 [19]. According to an online survey conducted during the early phase of the pandemic, more than 80% of the Japanese participants (n = 11,342) had implemented social distancing measures [2]. This shows that most Japanese people embraced preventative behavioral change and adhered to public health recommendations of wearing masks, hand hygiene, and social distancing since the early stage of the COVID-19 pandemic.

### Impact of refraining from physician visit for common infectious disease

The spread of COVID-19 limited clinical and laboratory diagnosis of common infectious diseases. In addition, people were unwilling to visit hospitals or clinics for diagnosis. Thus, the

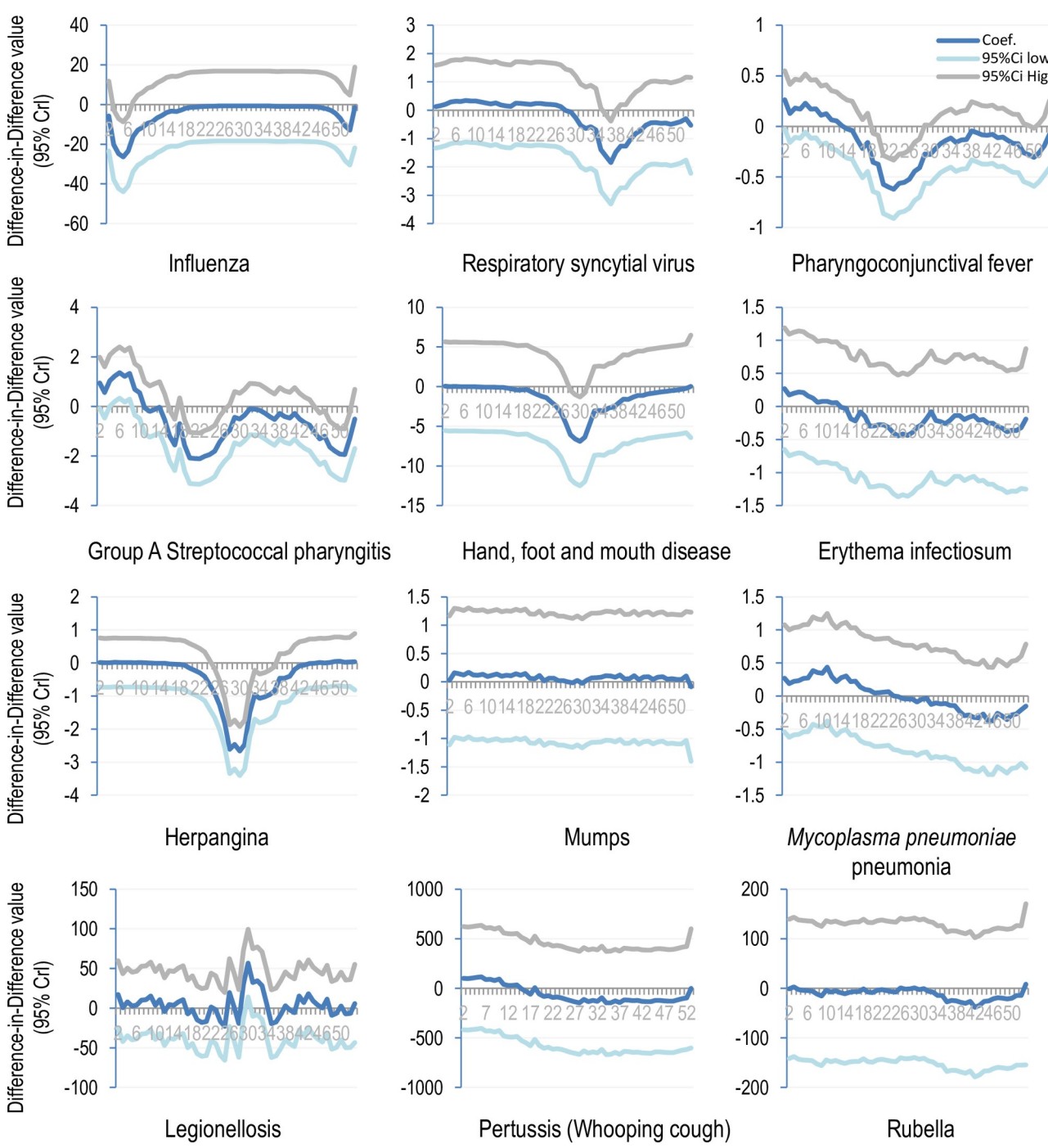

**Fig 4. Difference-in-differences value in 2020 vs. that in 2015–2019 (95% credible interval for droplet transmitted disease).** A negative 95% credible interval indicates fewer cases in the 2020 than in the 5 previous years (p<0.05). Ctrl: credible interval.

incidence of infectious diseases could be underreported. We showed the number of outpatients compared with that during the same month of the previous year at domestic medical clinics (Fig 5). Outpatient numbers declined in May and September 2020 in all departments. The decrease is remarkable, particularly in pediatrics. Epidemic weeks 14–21 (April 7–May 25) of 2020 were the period during which the Japanese experienced the first state of

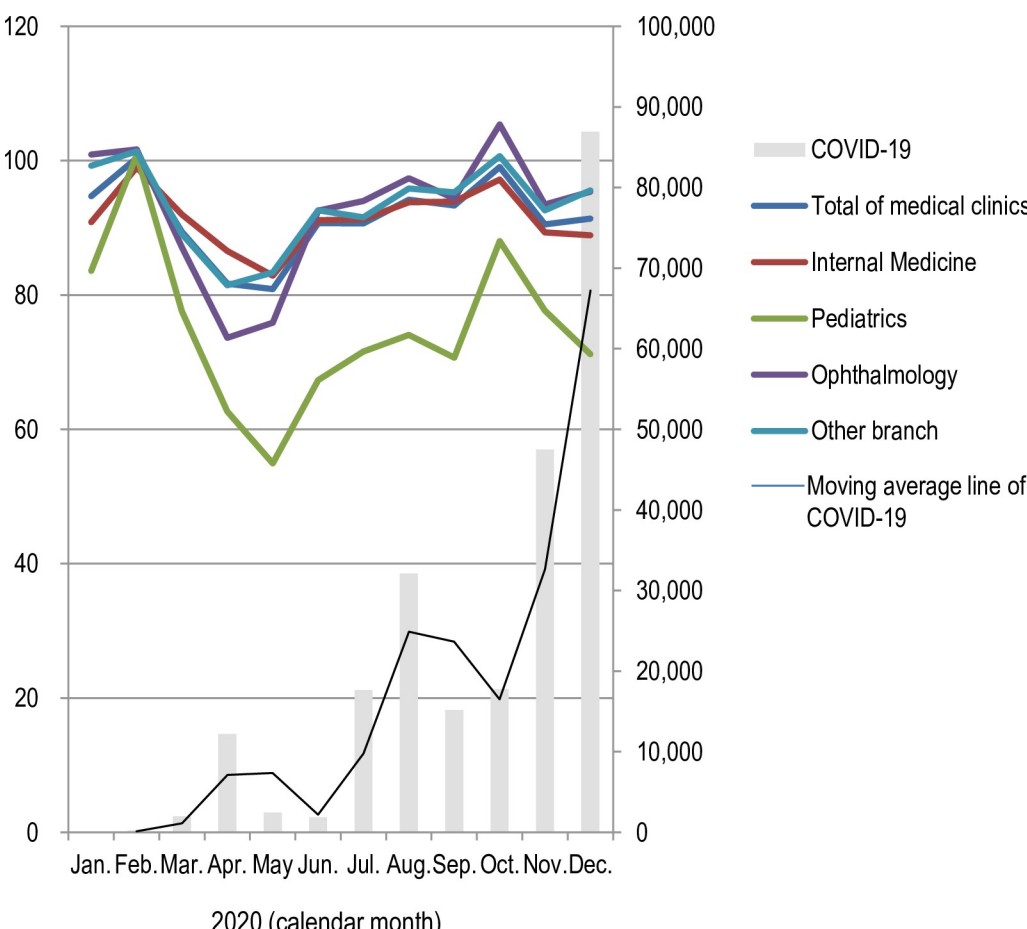

**Fig 5. Changes in the total number of outpatients that are shown compared with the same month last year.**

emergency. According to mobile phone location analysis, the number of people in major cities in Japan was reduced by 40–60% during the state of emergency [20]. Golden Week is one of major long vacation periods for Japanese. However, the Golden Week of 2020 (29 April—6 May 2020) was under the declaration of a state of emergency. On average, more than 50% of respondents spent the Golden Week period at home or in a neighborhood within a 3km radius of their home [21]. In other words, even the behavior of visiting hospitals and clinics may have been suppressed during this period. However, the decrease in outpatient numbers was temporary, and abstaining from visiting hospitals/clinics may have a minor impact. The effects of refraining from seeing a doctor may be considered negligible for infectious diseases with low numbers of cases through 1 year. Further, children who have a sudden fever or rash, such as measles or chickenpox, or patients who have high fever due to the flu may be less likely to refrain from seeing a doctor. It is often difficult for a citizen to distinguish COVID-19 from common infectious diseases. Therefore, even if individuals tended to abstain from visiting medical facilities, this had less effect on common infectious diseases that cause a high fever.

## Associations between COVID-19 measures and the incidence of droplet-transmitted diseases

The predominant transmission routes of SARS-CoV-2 are droplet and contact transmission [22]. Therefore, the Japanese government enacted public health measures to prevent droplet

and contact transmission. Measures such as wearing masks, sanitary hygiene, and avoiding the three Cs may have effectively prevented both COVID-19 and other droplet-transmitted and contact-transmitted diseases [23]. Notably, the number of patients with respiratory tract infections decreased during COVID-19 in Taiwan and Japan [23, 24]. The predominant transmission route of the influenza virus is droplet transmission, although the virus can also be transmitted through contact. A recent study showed that daily wearing of masks and hand hygiene are effective in preventing influenza transmissions in household settings [25]. The present study showed that the incidence of common infectious diseases classified in "droplet transmission" slowed in 2020. Influenza, pharyngoconjunctival fever, group A streptococcal pharyngitis, erythema infectiosum, *Mycoplasma pneumoniae* pneumonia, and pertussis showed epidemic trends before the COVID-19 outbreak but declined during the COVID-19 outbreak in Japan. Therefore, the prevention measures against COVID-19 might also be effective in preventing common infectious diseases transmitted through droplets [24].

*Mycoplasma pneumoniae* pneumonia is primarily a disease of school children (5–15 years of age). In Japan, schools were closed from week 9 to week 22. The incidence of *Mycoplasma pneumoniae* pneumonia declined from week 11. The incubation period of *Mycoplasma pneumoniae* pneumonia is approximately 2–3 weeks [26]. Considering this incubation period, the start of the fact that decline in infection rates was the third week after the school closure explains a lot. However, mycoplasma pneumonia is ubiquitous and active throughout the year, and many infections occur in household settings [26]. In addition, mycoplasma pneumonia is common in closed settings, such as summer camps or boarding schools. Due to school closure which minimizes close contact, the incidence of mycoplasma pneumonia declined in 2020. Pharyngoconjunctival fever is caused by adenovirus, a ubiquitous pathogen that can cause outbreaks in families and other closed settings, such as swimming pools or school camps [27]. Therefore, school closure may prevent the spread of pharyngoconjunctival fever. The closure of swimming pools and interruption of school camps was implemented in 2020. However, the hygiene intervention could not completely control the infection because adenovirus is highly infectious, and the virus is continually released by asymptomatic infected patients (about 30–35%) [28]. Epidemic keratoconjunctivitis, caused by adenovirus, also showed low activity during the COVID-19 outbreak in this study.

In contrast, school closure may not have effectively controlled group A streptococcal pharyngitis. The incidence of group A streptococcal pharyngitis showed a downward trend from week 10, and the lowest numbers were recorded in week 19. The numbers of cases increased from week 19 during the school closure. This trend has been observed in the last decade [8]. Group A streptococcal pharyngitis exhibits specific seasonality associated with school closure every year: the incidence increases during semesters and decreases during the long school breaks [29], although in the present study, the prolonged school closure due to COVID-19 did not result in the expected epidemic curve of group A streptococcal pharyngitis. The acute phase of group A streptococcal pharyngitis is the most infectious, and the acute phase infection rate is highest among siblings at 25% [29]. During the COVID-19 pandemic, children spent more time at home. Therefore, we presume that the incidence of group A streptococcal pharyngitis in mid-2020 reflects infection in the family setting.

In the present study, rubella and HFMD showed marked decrease throughout the year compared with 2015–2019. There may be several possible explanations for this. For example, in 2013, reported rubella cases were 14,344. However, a decline was noted in subsequent years, with 319 cases in 2014, 163 in 2015, 126 in 2016, and 93 in 2017 [30]. Since August 2018, there has been a rapid increase in rubella incidence: 2,941 in 2018 and 2,306 in 2019 [30]. The total number of rubella cases in 2020 was 100. It is inferred that the epidemic may have been contained and it is thought to have just returned to baseline, without the influence of COVID-19.

In this study, however, the CCF of rubella showed a negative correlation coefficient over lag 0–lag 20 weeks, suggesting that it might be suppressed over a long period. Incidentally, vaccination is effective in preventing rubella. However, in Japan, when routine vaccination of rubella was introduced, only girls were covered. Consequently, there are generations with low vaccination rates for rubella (males born between 1962 and 1978). The rubella epidemic of 2018–2019 mainly affected men in the indicated generations [30]. This suggests that Japan is still at risk for rubella epidemics in these susceptible populations. Therefore, the lower activity throughout the year of rubella may have been influenced by preventative behavioral changes associated with the COVID-19 epidemic. The causative agents of HFMD are enterovirus or coxsackievirus, and the epidemic often occurs in the summer, especially among children under 5 years of age [31]. Coxsackievirus A6 has been associated with large-scale epidemics since 2011 [31]. The large epidemic occurs every 2 years, and 2020 was the interval year. However, small epidemics caused by coxsackievirus A16 can occur during large-scale epidemics [31]. Viral transmission primarily occurs through droplets or contact. Therefore, handwashing and appropriate disposal of body waste are important for controlling the infection.

The present study showed that the mumps epidemic curve was stable. The patient count of mumps has been below 0.2 per sentinel site since 2018 and has been on a downward trend each year [8]. According to the present CCF analysis of mumps, the strongest negative correlation was obtained at lag 1 week. Vaccination against mumps has also become compulsory in many countries. However, the vaccination is still voluntary in Japan. According to the National Epidemiological Surveillance of Vaccine-Preventable Diseases, the vaccination coverage in recent years was 30~ 40% and the antibody coverage using sera stored in domestic serum banks was approximately 70% [32]. This means that there is a population that is susceptible to mumps in Japan. Therefore, the epidemic may have been controlled by the preventative behavioral change in this susceptible population.

This study showed that legionellosis had similar epidemic curves in 2015–2019 and 2020. Most infectious diseases caused by droplet transmission spread from person to person, although legionellosis is transmitted only from the water regime but does not spread from person to person [33].

## Associations between COVID-19 measures and the incidence of contact-transmitted diseases

In the present study, the incidence of epidemic keratoconjunctivitis, caused by adenovirus, was suppressed in 2020. Person-to-person transmission of adenovirus primarily occurs through contact with a patient's eyes, hands, or fomites [34]. Keratoconjunctivitis showed a negative correlation against COVID-19 in the lag from minus 20 weeks to 0 weeks. This may suggest that the hand-washing and hand-disinfection practices triggered by the COVID-19 epidemic also had a long-term effect on reducing the prevalence of contact infections.

However, the incidence of exanthema subitum and CRE in this study was not suppressed. Exanthema subitum is mostly caused by infection with human herpesvirus (HHV)-6 and, less frequently, HHV-7 β-herpesviruses in children aged below 2 years of age [35]. These viruses are ubiquitous and tend to infect infants during their first year of life [36]. More than 90% of children are seropositive by the age of 3 years [36]. The annual epidemic curve shows a minor change. Therefore, it is used as an index for the proper operation of pediatric sentinel site surveillance in Japan [37]. These findings may explain why exanthema subitum had similar epidemic trends in 2015–2019 and 2020.

CRE infects individuals across a wide age range in medical settings. Carbapenem-resistant strains arise from long-term use of carbapenem or broad-spectrum beta-lactams for

postoperative or other patients. The long-term use of such antibiotics is not directly associated with the outbreak of COVID-19. Therefore, a CRE epidemic was also observed in 2020.

## Associations between of COVID-19 measures and the incidence of airborne transmitted diseases

In the category of airborne transmission, only tuberculosis showed an epidemic. Surgical masks and daily hand hygiene cannot completely block the transmission of bacterial aerosols. Mass outbreaks of tuberculosis have been observed in crowded spaces or close-contact settings [38, 39]. According to a summary of mass outbreaks of tuberculosis from 2016 to 2018 in Japan, the most common location of infection was in business establishments (29.7–34.4%), followed by family and friends (18.9–24.4%), hospitals (15.6–17.8%), social welfare facilities (12.5–17.8%), and schools (12.5–24.3%) [40]. Therefore, the three Cs approach and remote work are expected to protect against the spread of airborne transmitted diseases, including tuberculosis. However, most of the recent tuberculosis cases in Japan are attributed to the reactivation of latent tuberculosis infection in older people [41]. Tuberculosis has a long latency period. Thus, the effects of the three Cs approach or lifestyle changes cannot be realized in a short period. Presumably, the COVID-19 outbreak has not affected the incidence of tuberculosis in 2020 [42]. However, the epidemic curve of tuberculosis in 2020 was relatively lower than that in 2019, as observed in Taiwan [43]. Komiya et al. (2020) reported that the number of laboratory tests for patients with suspected tuberculosis at the largest commercial laboratory in Japan in 2020 was significantly lower than in previous years. In 2020, medical checkups may have been canceled or delayed because of a fear of SARS-CoV-2 infection. But we have a different perspective. The proportion of foreign-born persons out of all TB cases was 11.08% in 2019, which among those aged among 15 and 39 years old has reached 61.6% in the same year [41]. It is probable that the case number of newly notified tuberculosis decreased due to the reduction in foreign workers and international students due to the restrictions of oversea travel imposed by COVID-19, and the decrease in employment opportunities for foreign residents [44, 45].

Similarly, chicken pox and measles can be transmitted to humans via aerosol, but they have shorter incubation periods than tuberculosis. Transmission of chicken pox and measles can be prevented by avoiding closed or crowded settings and wearing a mask. However, the marked reduction in measles may be explained by other factors. Measles and chicken pox can be prevented by vaccination; in particular, the measles vaccination rate is over 95% and the antibody prevalence is over 95% in all age groups among Japanese [46]. For this reason, Japan is now internationally recognized as a country that has eliminated measles, and this status has been maintained to date [46]. However, in 2019, the number of patients increased, especially due to an outbreak in a population that do not accept modern medicine, including vaccines, which spread to eight prefectures [46]. In addition, the outbreak of 2019 occurred in a commercial facility among patients with no or unknown vaccination history. The overall measles antibody prevalence in 2020 was 96.3%, although the antibody prevalence among children aged 1 year in 2020 was 69.8%, a significant decrease from 81.6% in 2019 [47]. This has been attributed to a temporary abstaining from vaccination [47]. However, the impact on the measles epidemic is thought to have been short-term and small. In addition, the estimated infected areas of the 12 measles cases reported in 2020 were 7 national, 4 overseas, and 1 unknown [47]. While the number of domestic cases has decreased, the proportion of imported cases is becoming more apparent. If there had been no entry restrictions due to the COVID-19 pandemic, such imported cases might have increased.

## Associations between of COVID-19 measures and incidence of fecal orally transmitted diseases

In the categories of fecal-oral transmission, infectious gastroenteritis caused by rotavirus did not show an epidemic trend. Rotavirus infections are common in children aged below 5 years, and transmission occurs through the fecal-oral route once the children touch an environment contaminated with stool [48]. Therefore, good hygiene, such as hand-washing and cleanliness, is important for the control of rotavirus. Among fecal-oral infections, only rotavirus infection showed a negative correlation between lag 0 and 25 weeks in the CCF analysis. This suggests that rotavirus infection occurs in the home rather than in restaurants and that appropriate hygiene behavior in the home may have had a long-term effect.

The Japan Food Service Association reported a drop in the overall restaurant sales of 17% in March and 40% in April compared with that in 2019. However, consumers purchased higher volumes of home-cooking-oriented products, such as baking products, pasta, and easy-to-prepare meal kits, during the nationwide state of emergency in Japan [49]. The demand for food delivery services or take-out services also increased during the COVID-19 pandemic [49]. Thus, the eating behaviors of people changed significantly from eating-out to eating at home or the workplace. The present study showed similar epidemic trends for EHEC in 2019 and 2020. In recent years, widespread sporadic cases of EHEC have occurred in relation to ready-to-eat food or catering meals [50]. For these reasons, the authors considered that enhanced hand-washing may have prevented person-to-person infection of EHEC but could not prevent food poisoning associated with ready-to-eat foods. The hepatitis A virus is transmitted via the fecal-oral route, primarily via contaminated water or food. Mass outbreaks of hepatitis A are often caused by the consumption of marine products at eating establishments in Japan [51, 52]. As previously mentioned, food consumption outside households has decreased during the COVID-19 pandemic [49]. The decline in the incidence of hepatitis A in 2020 compared with 2019 could be attributed to the reduced frequency of eating out. However, people can also eat marine products such as oysters or asari clams at home. Thus, household infection through such foods may be associated with sporadic occurrences. Recent evidence shows that the hepatitis A virus can be transmitted via sexual activity involving oral-anal contact [53]. In 2018, 926 cases of hepatitis A virus infection were reported in Japan according to NESID, and 349 cases (38%) were oral infections, whereas 396 cases (43%) were due to homosexual contact [54]. Pennanen-Iire et al. (2021) reported that psychological anxiety during the COVID-19 pandemic negatively impacted sexual activity [55]. Therefore, the decrease in hepatitis A incidence in 2020 may be due to the suppression of sexual contact.

Hepatitis E virus is transmitted via the consumption of uncooked deer or wild boar meat or fecally contaminated foods [56, 57]. However, recent domestic hepatitis E infections were caused by the consumption of raw pork meat or viscera [57]. Of 250 domestic cases from 2005 to November 2013 in Japan, 88 cases (35%) consumed pork meat/viscera, 60 cases (24%) consumed wild boar meat/viscera, 33 cases (13%) consumed deer meat/viscera, 10 cases (4.0%) consumed horse meat, and 11 cases (4.4%) consumed shellfish [58]. Notably, the infection rate and antibody prevalence of hepatitis E in Japanese domestic pigs are extremely high. The prevalence rate of anti-hepatitis E virus immunoglobulin G was 57% (n = 2242) and the prevalence rate of hepatitis E virus RNA was 7% (n = 192) following the analysis of serum samples obtained from 3,925 edible pigs from 117 farms across 21 prefectures [57]. The Japanese government banned the serving of raw meat at eating establishments in June 2015 [59]. Therefore, the main areas where raw meat/viscera is consumed are presumably households, and the influence of the COVID-19 pandemic on hepatitis E incidence was low. However, the number of patients decreased from week 23, 2020, to week 39, 2020, compared with that during the

previous year. As the average incubation period of hepatitis E virus is 6 weeks (28–60 days) [60] and the nationwide state of emergency was effected from week 16 to week 22 in 2020, the declining HEV incidence during the summer season was possibly due to behavior regulation and the long incubation period of HEV.

## Associations between of COVID-19 measures and the incidence of vector-borne diseases

Vector-borne diseases had larger epidemic curves in 2020 than in 2015–2019. For scrub typhus, all cases (n = 76) in 2018 and all cases (n = 397) in 2019 occurred in domestic settings [61]. Scrub typhus is a rickettsiosis caused by *Orientia tsutsugamushi* and is transmitted to people through bites by infected chiggers (larval mites) [62]. People are bitten by chiggers during agricultural work or outdoor activities. Japanese people moved to rural areas to escape COVID-19 in urban areas, and outdoor activities became increasingly popular in Japan during the COVID-19 pandemic [63]. The implication was a higher risk of infection with scrub typhus. In this study, we also examined the CCFs between the two-time series of scrub typhus and COVID-19 disease counts. The peak incidence of COVID-19 coincided with the peak incidence of scrub typhus exhibiting a 1-week lag. Although there is a large seasonal bias in the occurrence of scrub typhus disease, we cannot deny the possibility that the mobility change of people by COVID-19 epidemic led to an increase in the incidence of scrub typhus infection.

## Associations between of COVID-19 measures and the incidence of STDs

The survey showed a similar epidemic curve for AIDS and syphilis in 2020, although the overall incidence of AIDS and syphilis was slightly lower than that of 2015–2019. Some studies have investigated the effect of COVID-19 on the incidence of STDs. The implementation of social confinement measures reduced the incidence of syphilis [64, 65]. Consequently, the relaxation of social confinement measures increased the incidence of syphilis due to the resumption of sexual activities [64]. However, such social regulations were relatively loose in Japan, and residents could act freely, even in a state of emergency. This situation in Japan might be similar to that in Taiwan and Spain [66, 67]. Taiwan did not implement a state of emergency, instead implementing border controls and infection prevention measures from an early stage. The total number of syphilis cases diagnosed in Taiwan was 6,294 in 2019 and 5,776 in 2020 [66]. The total number of human immunodeficiency virus (HIV) cases diagnosed in Taiwan was 1,000 in 2020 and 1,293 in 2019. Chia et al. (2020) suggested that the reduction in STDs during the COVID-19 pandemic might reflect less frequent sexual encounters due to the fear of SARS-CoV-2 infection [66]. According to the national online database by YouGov involving three countries on September 21, 2020, Japanese people (72%) were most worried about contracting COVID-19, followed by Spanish people (57%) and Taiwanese people (53%) [68]. This may explain the decline in the total number of AIDS and syphilis cases. However, further explanations of additional underlying factors are needed. According to the AIDS Trends Committee, the number of HIV antibody tests nationwide in the second quarter of 2020 was 0.27 times that of the same quarter of the previous year (9,584 vs. 3,5908) [69]. Thus, the decrease in the number of antibody tests might explain the decrease in the number of new patients with syphilis and HIV [62, 63].

The epidemic curve of amoebiasis in 2020 was lower than in 2015–2019. Amoebiasis can be transmitted to humans orally after touching infected feces, or eating or drinking food or water that is contaminated with the parasite [70]. Among the 9,301 cases reported by NESID during 2007–2016, infections attributed to oral consumption accounted for 22%, and sexual transmission accounted for 29% of all cases [71]. The effects of the COVID-19 pandemic on sexually

transmitted amoebiasis are similar to those discussed for other STDs. However, for fecal-oral amoebiasis infections, hygienic interventions for the COVID-19 pandemic may have reduced the number of people infected with amoebiasis.

The lowest peak of current incidents of sexual transmitted disease showed a lag of few weeks than the peak of COVID-19 by cross-correlation analysis. The results may support the view that people became fearful when they saw the rapid increase in the number of new COVID-19 cases and changed their behavior.

## Limitation of this study

Our study has some limitations and should be interpreted with caution. It will be necessary to adjust age and sex for the incidence of each infectious disease [72]. However, as for influenza shown in Fig 2, since the number of definitive diagnoses continues to be extremely small in 2020, it is impossible to consider age and sex. For this reason, we judge that it is important to understand the overall picture of each disease rather than considering age and sex. For respiratory syncytial virus infections, it is especially important to consider age, and there are other outbreaks in some regions in Japan that eventually spread nationwide in 2021 [73]. Regarding respiratory syncytial virus infections, the authors would like to perform the analysis with age and sex as factors in future studies. The present study did not evaluate regional differences, and further detailed studies are warranted to investigate this issue. By investigating countries where people are less afraid of COVID-19 [68], or a country such as Taiwan where the division of roles between laboratory testing institutions is clear, it may be possible to eliminate the effects of refraining from physical visits or clarify the impact on laboratory tests for common infectious diseases. In addition, regional differences can be eliminated by comparing regions with similar environments.

In conclusion, our study demonstrated the potential impact of public health interventions for the COVID-19 pandemic in preventing the transmission of droplet-transmitted infectious diseases and those transmitted through other routes. After the COVID-19 pandemic, daily hygienic behavior, avoidance of the three Cs, wearing masks in public spaces, and remote work might become the global new normal.

## Supporting information

**S1 Fig. Cross-correlation graph.** S1A: Cross-correlation between COVID-19, and common infectious diseases under the national sentinel surveillance. S1B: Cross-correlation between COVID-19, and common infectious diseases under the national notifiable disease surveillance. The X-axis indicates the correlation coefficient between COVID-19 and each common infectious disease. The Y-axis indicates lag (week).
(PPTX)

**S1 Table. Case numbers per week about COVID-19 and case number per week per sentinel cite of notifiable infectious diseases.** These weekly numbers of COVID-19 were calculated from the daily number of the new positive cases that have taken polymerase chain reaction for SARS-CoV-2 or antigen testing for SARS-CoV-2. The data was obtained from the Ministry of Health, Labour and Welfare [7]. The numbers of notifiable infectious diseases were collected from the Infectious Diseases Weekly Reports in accordance with the National Epidemiological Surveillance of Infectious disease [8].
(XLSX)

**S2 Table. The total number of out-of-hospital (outpatient and home medical care) cases in the medical clinics and insurance pharmacies compared with that in the same month of**

**the previous year.** The number of cases referred to here is the number of medical fee statements (receipts), and each medical institution prepares one statement for one patient every month. The data is national data and are obtained from the Ministry of Health, Labor and Welfare's estimated medical expenses database [11].
(XLSX)

## Acknowledgments

The authors are grateful to the two reviewers and the editor for comments and suggestions. We would like to thank Editage for English language editing.

## Author Contributions

**Data curation:** Kenji Hibiya, Akira Shinzato, Shinichiro Ueda.

**Formal analysis:** Hiroyoshi Iwata.

**Investigation:** Kenji Hibiya.

**Methodology:** Takeshi Kinjo, Masao Tateyama.

**Supervision:** Masao Tateyama, Shinichiro Ueda, Jiro Fujita.

**Validation:** Takeshi Kinjo.

**Writing – original draft:** Kenji Hibiya.

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
