## [Decision Letter · Decision Letter 0]

14 Jul 2021

PONE-D-21-11519

Activity of common infectious diseases during the COVID-19 pandemic

PLOS ONE

Dear Dr. Hibiya,

Thank you for submitting your manuscript to PLOS ONE. After careful consideration, we feel that it has merit but does not fully meet PLOS ONE’s publication criteria as it currently stands. Therefore, we invite you to submit a revised version of the manuscript that addresses the points raised during the review process.

We look forward to receiving your revised manuscript.

Kind regards,

Martial L Ndeffo Mbah, Ph.D

Academic Editor

PLOS ONE

Journal Requirements:

 [No].

3. Please include your tables as part of your main manuscript and remove the individual files. Please note that supplementary tables (should remain/ be uploaded) as separate "supporting information" files.

Reviewers' comments:

Reviewer's Responses to Questions

**Comments to the Author**

1. Is the manuscript technically sound, and do the data support the conclusions?

Reviewer #1: Yes

Reviewer #2: Partly

2. Has the statistical analysis been performed appropriately and rigorously? 

Reviewer #1: Yes

Reviewer #2: N/A

3. Have the authors made all data underlying the findings in their manuscript fully available?

Reviewer #1: No

Reviewer #2: Yes

4. Is the manuscript presented in an intelligible fashion and written in standard English?

Reviewer #1: Yes

Reviewer #2: Yes

5. Review Comments to the Author

Reviewer #1: My comments (in no particular order) are focused primarily on the authors’ methods and presentation.

1. The graphs indicate “week” along the x-axis. Does this represent calendar week

(i.e., starting from January 1) or “seasonal” time (e.g., one might take influenza season as extending from July to June).

2. It is not at all clear to me how one can observe fractional numbers of cases per week

(e.g., as with respiratory syncytial virus).

3. The authors present counts from (A) national sentinel surveillance (e.g., Figure 2A) and (B) national notifiable disease (e.g., Figure 2B). Since numbers are much smaller for the former versus the latter, the question arises how well A tracks B. Are there any common diseases to A and B, and if so, are the patterns similar?

4. The authors might formalize the relationship (if any) between disease counts and COVID-19 incidence by examining the cross correlation function between the two time series. It would be of interest if lags could then be identified.

5. One might more readily be able to identify trends or periodicities in the graphs if the authors were to smooth the incidence curves (e.g., moving averages). Given counts from national sentinel surveillance are typically quite small (in the single digits), eliminating to some extent random fluctuations might be useful.

6. In a similar spirit, would it be useful to combine diseases in Figure 3 according to mode of transmission? Would a clearer pattern thereby emerge?

7. Why counts and not rates? And, should not there be age adjustments, or gender considerations? For example, the “usual” U-shaped incidence curve for influenza is lost by pooling across all ages.

Reviewer #2: In this manuscript, the authors aim to evaluate whether restrictions and interventions in place during COVID-19 in Japan were associated with decreases in reported cases of other infectious diseases. They compare total case counts between the years 2019 and 2020, as well as epidemic curves for weekly cases, and find that several pathogens, particularly those transmitted by the droplet route, had strikingly lower numbers of reported cases during the COVID-19 epidemic in 2020. They also include a very thorough discussion of their results for various pathogens, including results that were unexpected, which I appreciated.

Overall, the comparison of epidemic curves, and total cases, between 2019 and 2020 support the overall conclusion that the measures taken to curb the spread of COVID-19 may have also prevented the spread of certain infections. My main comments here are (1) whether a statistical analysis can be used to quantify weekly changes in case counts as well as to tie the story together a bit more clearly, and (2) how much of the reduction in these other infections might be attributable to reduced healthcare seeking and/or reduced capacity to test for other pathogens during COVID-19. Please find these and additional comments described in detail below.

Main comments:

1) I suggest using a difference-in-difference regression model to statistically evaluate changes in weekly case counts for each of these pathogens. While I very much appreciate the detailed discussion of the trends for each pathogen, and I agree that the data speaks for itself in many of the examples, I think a quantitative analysis such as this would help pull the manuscript together with a clearer story. This would also help to identify which week(s) during 2020 each pathogen had significantly different case counts from the previous year(s). Examples of this approach can be found in the Sakamoto et al. (2020) JAMA analysis of changes in influenza epidemics in Japan, as well as in the Lee & Lin (2020) EID study on effects of COVID-19 on infections in Taiwan.

2) The authors have very thoughtfully extracted and categorized the reported case data. Is any data additionally available that could be used to examine the impact of reduced healthcare seeking or testing capacities on the conclusions? If possible, this should be investigated further. For example, is information available on changes in number of tests performed or in number of hospital admissions? If so, this could be used to perform a sensitivity analysis of the total 2020 case counts followed by the regression-based analysis described above (see Lee & Lin (2020) EID for an example). If this is not possible, then this potential issue should be discussed in more depth and earlier on in the manuscript (not just in the limitations and discussion of HIV/syphilis). For example, are there certain pathogens this will likely affect more than others in Japan? Might this be different in the sentinel vs. passive surveillance systems?

3) Why was the year 2019 alone used as the comparison for 2020? Why not compare several previous seasons, for example 2014-2019 with 2020, to account for some of the year-to-year variation?

Additional comments:

• In the abstract, it is unclear what methods and results the authors used to arrive at the conclusions. Additional details and using a structured abstract may help to improve clarity. Including the statistical analysis suggested above may also help in clarifying what the key findings are in the abstract.

• I don’t understand what the sentence on lines 68-69 is meant to indicate. Please rephrase or expand here.

• In the methods, more details are needed on why the inclusion/exclusion criteria were applied. Why was 400 chosen as the minimum number of cases? Why were fulminant, invasive, and enteric infections other than rotavirus excluded?

• I suggest using “common infectious diseases” rather than “representative infectious diseases” throughout the manuscript. If “representative” is meant to indicate something other than “common”, please explain further.

• I suggest using the phrase “fecal-oral transmission” instead of “oral transmission”.

• In Figure 1, I suggest using the y-axis label “Number of cases per day”.

• In Figures 2-3, please use colorblind-friendly colors instead of green and red.

6. PLOS authors have the option to publish the peer review history of their article (what does this mean?). If published, this will include your full peer review and any attached files.

Reviewer #1: **Yes: **James A. Koziol

Reviewer #2: No

---

## [Author Response · Author response to Decision Letter 0]

15 Sep 2021

5. Review Comments to the Author

Reviewer #1: My comments (in no particular order) are focused primarily on the authors’ methods and presentation.

1. The graphs indicate “week” along the x-axis. Does this represent calendar week (i.e., starting from January 1) or “seasonal” time (e.g., one might take influenza season as extending from July to June).

Reply to the reviewer: The week on the horizontal axis of Figure 2 and Figure 3 indicates an epidemiological week (for example, the first week in 2020 is from December 30 to January 5, and the first week in 2019 is from December 31 to January 6). The definition of weeks was based on the “Report week correspondence table” (see https://www.niid.go.jp/niid/ja/calendar.html) presented by the National Institute of Infectious Diseases. Although the definition of epidemiological weeks varies from year to year, but the difference is only a few days, not large enough to affect over the months.

We added the explanation about the “week” in the main text. 

Material and Methods (insert on page 7, line 91)

Though the actual reported numbers were reported per week, this “week” indicates an epidemiological week. For example, the first week in 2020 is from December 30 to January 5, and the first week in 2019 is from December 31 to January 6. This is referred to as the “weeks ending log” prescribed by the National Institute of Infectious Diseases (see https://www.niid.go.jp/niid/en/calendar-e.html). 

Description of Fig 2 (insert on page 12, line 148 )

The “week” along the x-axis indicates the epidemiological week.

2. It is not at all clear to me how one can observe fractional numbers of cases per week

(e.g., as with respiratory syncytial virus).

3. The authors present counts from (A) national sentinel surveillance (e.g., Figure 2A) and (B) national notifiable disease (e.g., Figure 2B). Since numbers are much smaller for the former versus the latter, the question arises how well A tracks B. Are there any common diseases to A and B, and if so, are the patterns similar?

Reply to the reviewer: We do apologize for the confusion, as the fractional numbers of cases per week were unique to Japan and were difficult to understand. In Japan, fractional numbers of cases per week surveys are conducted for common diseases, and 100% of surveys are conducted for important diseases for which it is judged how all cases should be grasped. Figure 2A shows survey results of the fractional numbers of cases per week. Table 1 shows the number of facilities for each fractional numbers of cases per week. The vertical axis is the number of cases diagnosed at each facility per week divided by the number of facilities. Since it is difficult to understand the expression method on the vertical axis, we will change it to the expression method we have used in our previous articles as follows: The vertical bar represents the number of patients referred to one hospital per week per selected time point. On the other hand, Figure 2B shows a 100% survey. Since the fractional number of cases per week survey or the 100% survey is decided depending on the disease, these data cannot be duplicated.

Material and Methods (revised the sentences on page 5, line 80-82)

Sentinel surveillance systems involve a network of reporting sites (sentinel sites), including doctors, laboratories, or public health departments. Public health departments receive data from each sentinel sites (medical care facilities).

➡

Sentinel surveillance systems involve clinics or hospitals or public health centers, local infectious disease surveillance centers (local IDSCs), and national infectious disease surveillance centers (national IDSC). The public health center gathered a total number of patients during one week with target diseases diagnosed at each medical facility with influenza, pediatric, ophthalmic, and designated sentinel sites. Local IDSCs gather the data from public health centers in a prefecture. Finally, national IDSC gather the data from local IDSCs. The weekly number of cases includes number of cases diagnosed at each facility per week divided by the number of facilities with sentinel sites.

Material and Methods (We revised sentence of the page 5, line 81-82)

Public health departments receive data from each sentinel sites (medical care facilities).

➡

The public health center gathered a total number of patients during one week with target diseases diagnosed at each medical facility with influenza, pediatric, ophthalmic, and designated sentinel sites. Local IDSCs gather the data from public health centers in a prefecture. Finally, national IDSC gather the data from local IDSCs. The weekly number of cases includes number of cases diagnosed at each facility per week divided by the number of facilities with sentinel sites. The weekly number of reports is calculated by dividing the weekly number of patient reports by the number of medical institutions with sentinel sites. This number of sentinel sites is to reflect Japan’s overall trends regarding infectious disease epidemics. Sentinel sites are set under the jurisdiction of the public health center so that the morbidity rate in nationwide can be estimated with a standard error rate of ≤5%.

Legends of Table 2 (Inserted to page 10, line 107)

The vertical bar represents the number of patients referred to one medical facility per week per selected time point, but italic figures show the total number by 100% survey.

Figure legends (revised the sentence in line 143 of page 12 / inserted on page 15, line 198)

The vertical axis represents the number of patients referred to one medical facility per week per selected time point.

4. The authors might formalize the relationship (if any) between disease counts and COVID-19 incidence by examining the cross correlation function between the two time series. It would be of interest if lags could then be identified. 

Reply to the reviewer: We examined the cross-correlation function between the two-time series of COVID-19 and common infectious diseases count and described it in the Results section. In addition, cross-correlogram of each disease are shown as supplemental figures. We added statistics analysis in the Material and Methods section as follows: 

Material and Method

Statistical analysis (Inserted on page 10, line 108)

We examined the cross-correlation functions (CCF) between the two-time series of common infectious disease count and COVID-19 disease count. We used the CCF to understand the time-lagged correlation between common infectious disease count and COVID-19 incidence. For the statistical analysis, Light Stone® STATA® ver.15 was used.

Results (Inserted on page 14, line 191)

Additionally, we examined the CCF between the two-time series of COVID-19 and common infectious disease counts. Overall, there were no diseases showing a strong correlation with COVID-19 (S1 Fig). Among them, cross-correlations in Scrub typhus were the highest at rag –1 week (CCF 0.8075), and cross-correlations in herpangina were the highest at rag –15 week (CCF 0.6465).

5. One might more readily be able to identify trends or periodicities in the graphs if the authors were to smooth the incidence curves (e.g., moving averages). Given counts from national sentinel surveillance are typically quite small (in the single digits), eliminating to some extent random fluctuations might be useful.

Reply to the reviewer: Thank you for your advice. The moving average lines have been added to the column graph of COVID-19 and the incidence curve of common infectious disease. We append this to the section of Materials and Methods as follows;

Material and Methods

Figure legend (Inserted on page 12, line 147)

The moving average lines were applied to the column graph of COVID-19 incidence (gray tick line) and to the incidence curve of 2020 (scarlet tick line) of each common infectious disease. The “week” along the x-axis indicates the epidemiological week.

Figure legend (Inserted on page 15, line 198)

The moving average lines were applied to the incidence curve of 2020 (scarlet tick line) and 2015-2019 (blue tick line) of each common infectious disease.

6. In a similar spirit, would it be useful to combine diseases in Figure 3 according to mode of transmission? Would a clearer pattern thereby emerge?

Reply to the reviewer: Thank you very much for your comment that each disease in Figure 3 should be combined. However, if you look closely at Figure 3, there is a big difference for each disease, especially for oral infections. As suggested by the reviewer, we also think that it is important to consider each infection route. However, since there are differences for each disease, and we believe that it is more accurate to show individual data.

7. Why counts and not rates? And, should not there be age adjustments, or gender considerations? For example, the “usual” U-shaped incidence curve for influenza is lost by pooling across all ages.

Reply to the reviewer: Thank you for your important suggestions. We have already responded to Japan's unique surveys on fractional numbers of cases per week. As for the influenza that you pointed out, as shown in Figure 2, since the number of definitive diagnoses continues to be extremely small in 2020, it is impossible to consider age and gender. For this reason, we judge that it is important to understand the overall picture of influenza rather than considering age and gender. For respiratory syncytial virus infections, it is especially important to consider age, and there are limited epidemics in some regions in Japan in 2021. Regarding respiratory syncytial virus infections, the authors would like to perform the analysis considering age and gender in future studies.

Discussion

Limitation of this study (Inserted on page 29, line 474)

It will be necessary to adjust age and gender for the incidence of each infectious disease [69]. However, as for the influenza shown in Figure 2, since the number of definitive diagnoses continues to be extremely small in 2020, it is impossible to consider age and gender. For this reason, we judge that it is important to understand the overall picture of each disease rather than considering age and gender. For respiratory syncytial virus infections, it is especially important to consider age, and there are limited epidemics in some regions in Japan in 2021 [70]. Regarding respiratory syncytial virus infection, the authors would like to perform the analysis considering age and gender in future studies.

References (Added new citation)

69. Galasso V, Pons V, Profeta P, Becher M, Brouard S, Foucault M. Gender differences in COVID-19 attitudes and behavior: Panel evidence from eight countries. Proc Natl Acad Sci U S A. 2020;117: 27285-27291. 

70. Ujiie M, Tsuzuki S, Nakamoto T, Iwamoto N. Resurgence of Respiratory Syncytial Virus Infections during COVID-19 Pandemic, Tokyo, Japan. Emerg Infect Dis. 2021;27. 

8. My revision

We needed a reference for the statistical analysis in this revision. Therefore, I added two authors for this paper.

Hiroyoshi Iwata, Clinical Pharmacology & Therapeutics, University of the Ryukyus School of Medicine 

Shinichiro Ueda, Clinical Pharmacology & Therapeutics, University of the Ryukyus School of Medicine

We changed “Varicella” to “Chicken pox” because the name of “Chicken pox” is more common in children. Similarly, pertussis is sometimes used as the name of “Whooping cough;” thus, it is shown in parentheses. 

In this revision, we added supplemental table and figure. Therefore, I described the supporting information.

Supporting information

S1 Fig. Cross-correlation graph. S1A: Cross-correlation between COVID-19, and common　infectious diseases under the national sentinel surveillance. S1B: Cross-correlation between COVID-19, and common infectious diseases under the national notifiable disease surveillance. The X-axis indicates the correlation coefficient between COVID-19 and each common infectious disease. The Y-axis indicates Lag (week).

S1 Table. Case numbers per week about COVID-19 and case number per week per sentinel cite of notifiable infectious diseases. These weekly numbers of COVID-19 were calculated from the daily number of the new positive cases that have taken polymerase chain reaction for SARS-CoV-2 or antigen testing for SARS-CoV2. The data was obtained from the Ministry of Health, Labour and Welfare [7]. The numbers of notifiable infectious diseases were collected from the Infectious Diseases Weekly Reports in accordance with the National Epidemiological Surveillance of Infectious disease [8].

S2 Table. The total number of out-of-hospital (outpatient and home medical care) cases in the medical clinics and insurance pharmacies compared with that in the same month of the previous year. The number of cases referred to here is the number of medical fee statements (receipts), and each medical institution prepares one statement for one patient every month. The data is national data and are obtained from the Ministry of Health, Labor and Welfare's estimated medical expenses database [11].

 

Reviewer #2: In this manuscript, the authors aim to evaluate whether restrictions and interventions in place during COVID-19 in Japan were associated with decreases in reported cases of other infectious diseases. They compare total case counts between the years 2019 and 2020, as well as epidemic curves for weekly cases, and find that several pathogens, particularly those transmitted by the droplet route, had strikingly lower numbers of reported cases during the COVID-19 epidemic in 2020. They also include a very thorough discussion of their results for various pathogens, including results that were unexpected, which I appreciated.

Overall, the comparison of epidemic curves, and total cases, between 2019 and 2020 support the overall conclusion that the measures taken to curb the spread of COVID-19 may have also prevented the spread of certain infections. My main comments here are (1) whether a statistical analysis can be used to quantify weekly changes in case counts as well as to tie the story together a bit more clearly, and (2) how much of the reduction in these other infections might be attributable to reduced healthcare seeking and/or reduced capacity to test for other pathogens during COVID-19. Please find these and additional comments described in detail below.

Main comments:

1) I suggest using a difference-in-difference regression model to statistically evaluate changes in weekly case counts for each of these pathogens. While I very much appreciate the detailed discussion of the trends for each pathogen, and I agree that the data speaks for itself in many of the examples, I think a quantitative analysis such as this would help pull the manuscript together with a clearer story. This would also help to identify which week(s) during 2020 each pathogen had significantly different case counts from the previous year(s). Examples of this approach can be found in the Sakamoto et al. (2020) JAMA analysis of changes in influenza epidemics in Japan, as well as in the Lee & Lin (2020) EID study on effects of COVID-19 on infections in Taiwan.

Reply to the reviewer: We applied to our data a difference-in-differences regression model to statistically evaluate change in weekly case counts for each of the infectious diseases. We then described the statistical analysis in the Material and Methods and Results sections as follows:

Material and Methods

Statistical analysis (Inserted on page 10, line 108)

We examined the cross-correlation functions (CCFs) between the two-time series of common infectious disease count and COVID-19 disease count. We used the CCF to understand the time-lagged correlation between common infectious disease count and COVID-19 incidence. To compare the change in case numbers of common infectious diseases in 2020 with that in the previous 5 years (2015–2019), a difference-in-differences linear regression was applied for infectious diseases transmitted by droplets that are most susceptible to behavioral changes caused by COVID-19. The model included a categorical variable for each week, a categorical variable for the 2020 season (versus the 2015–2019 seasons) and the interaction variables for each week and the 2020 season, following the method described by Sakamoto et al. (2020) [5]. For the statistical analysis, Light Stone® STATA® ver.15 was used.

Results (Inserted on page 16, line 219)

According to the difference-in-differences analysis, the activity of influenza was significantly lower since the second week in 2020 than during 2015-2019 (Fig 4). Similarly, respiratory syncytial virus was lower after 27 weeks, Group A streptococcal pharyngitis was lower after 10 weeks, hand, foot, and mouth disease was lower after 5 weeks, erythema infectiosum were lower after 13 weeks, herpangina was lower after 11 weeks, mumps was lower during 27 to 30 weeks, Mycoplasma pneumoniae pneumonia was lower after 25 weeks, pertussis was lower after 15 weeks, and rubella was lower after 31 weeks (Fig 4). However, legionellosis was more frequent throughout the year than in 2015–2019 (Fig 4).

Figure legend (Page 16, line 219)

Fig. 4 Difference-in-differences value in 2020 vs. that in 2015-2019 (95% credible interval for droplet transmitted disease). Negative 95% credible interval indicates fewer cases in the 2020 than in the 5 previous years (p<0.05). Ctrl: credible interval 

2) The authors have very thoughtfully extracted and categorized the reported case data. Is any data additionally available that could be used to examine the impact of reduced healthcare seeking or testing capacities on the conclusions? If possible, this should be investigated further. For example, is information available on changes in number of tests performed or in number of hospital admissions? If so, this could be used to perform a sensitivity analysis of the total 2020 case counts followed by the regression-based analysis described above (see Lee & Lin (2020) EID for an example). If this is not possible, then this potential issue should be discussed in more depth and earlier on in the manuscript (not just in the limitations and discussion of HIV/syphilis). For example, are there certain pathogens this will likely affect more than others in Japan? Might this be different in the sentinel vs. passive surveillance systems?

Reply to the reviewer: We showed the number of monthly hospital visitors of 2020 as Figure 5 with the number of monthly newly infected COVID-19 cases. Unfortunately, the number of outpatients for each disease cannot obtain from the national data. Therefore, we discussed the potential issue and further utilized the data on the manuscript as follows:

Discussion (Inserted to page17, line236)

Impact of refraining from physician visit for common infectious disease

The spread of COVID-19 limited the clinical and laboratory diagnosis of common infectious diseases. In addition, people were unwilling to visit hospitals or clinics for diagnosis. Thus, the incidence of infectious diseases could be underreported. We showed the number of outpatients compared with that during the same month of the previous year at domestic medical clinics (Fig. 5). Outpatient numbers declined in May and September 2020 in all departments. Especially in pediatrics, the decrease is remarkable. Epidemic weeks 14–21 (April 7–May 25) of 2020 were the period during which the Japanese experienced the first state of emergency. According to mobile phone location analysis, the number of people in major cities in Japan was reduced by 40–60% during the state of emergency [20]. On average, the ratio of people who spent their time at home or in the neighborhood within a radius of 3 km from their homes had shown a decrease of over 50% during a long vacation (April 29 to May 6, 2020) under of state of emergency [21]. In other words, even the behavior of visiting hospitals and clinics may have been suppressed during this period. However, the decrease in outpatient numbers was temporary, and abstaining from visiting hospitals/clinics may have a minor impact. At least, for infectious diseases with low numbers of cases through 1 year, the effects of refraining from seeing a doctor may be considered to be negligible. Also, children who have a sudden fever or rash, such as measles or chickenpox, or patients who have high fever due to the flu may be less likely to refrain from seeing a doctor. It is often difficult for a citizen to distinguish COVID-19 from common infectious diseases. Therefore, even if individuals tended to abstain from visiting medical facilities, we suspect it had less effect on common infectious diseases that cause a high fever.

Material and Methods

Total number of outpatient cases in the medical clinics (Inserted on page 10, line 108) 

The total number of outpatient cases in the domestic medical clinics was obtained from the Ministry of Health, Labor and Welfare's estimated medical expenses database [11]. The “medical clinic” is meaning the places where doctors provide medical practice and do not have hospitalization facilities for patients or have hospitalization facilities for 19 or less patients. The number of cases referred to here is the number of medical fee statements (receipts), and each medical institution prepares one statement for one patient every month. The data was added as S2 Table.

References (Added a new reference cited in the above description)

11. Medical expenses data. The Ministry of Health, Labor and Welfare. [Cited 2021 August 31]. Available from: https://www.mhlw.go.jp/bunya/iryouhoken/iryouhoken03/03.html (see S2 Table)

20. COVID-19 Mobility Trends Reports-Apple, Apple Maps. [Cited 2021 September 3]. Available from: 

https://covid19.apple.com/mobility

21. Decrease rate of visitors in each prefecture during Golden Week. LocationMind xPop [Cited 2021

 September 3]. Available from: https://locationmind.com/news/262/ (Japanese)

We added the explanation about the Fig 5 in the Results section.

Results (Inserted on page 16, line 219)

Relationship between the COVID-19 epidemic and total number of outpatient cases. 

The total number of outpatients is shown year-on-year in the same month (Fig 5, S2 Table). At the same time, the epidemic curves of COVID-19 were superimposed. The number of outpatients in any clinical departments decreased in May and September. The decrement was greatest in pediatrics. The transition was similar to the COVID-19 epidemic curve. We tried to calculate the cross-correlation coefficient between them, but could not analyze it for unknown reasons. 

3) Why was the year 2019 alone used as the comparison for 2020? Why not compare several previous seasons, for example 2014-2019 with 2020, to account for some of the year-to-year variation?

Reply to reviewer: In Figure 2, the 2019 epidemic curve was used to show the connection from the previous year to the current disease epidemic. If the number of patients has decreased from the beginning of 2020 compared to the same month of the previous year, it is possible to know when it started last year　However, in Figure 3, we showed the average value from 2015 to 2019 as the comparison for 2020.

Table legend of Fig 3 (revised the sentence, P15L197-198): 

The green lines (-) indicate the 2019 epidemic curves.

➡

The blue thin lines (-) indicate the epidemic curves during 2015-2019. 

Additional comments:

• In the abstract, it is unclear what methods and results the authors used to arrive at the conclusions. Additional details and using a structured abstract may help to improve clarity. Including the statistical analysis suggested above may also help in clarifying what the key findings are in the abstract.

Reply to reviewer: We included the statistical analysis suggested by reviewers in this study. Then, we revised the abstract as follows: 

Abstract (revised the sentences in page 2, line 33 to 43)

When the cross-correlation functions (CCFs) between the two-time series of common infectious disease count and COVID-19 disease count were examined, a strong correlation was shown for scrub typhus at rag -1 week (CCF 0.8705), herpangina at rag -15 week (CCF 0.6465). The overall infectious activity of droplet-transmitted diseases in 2020 was lower than the average of previous five years (2015–2019). According to the difference-in-differences analysis, the activity of influenza and rubella was significantly lower since the second week in 2020 than those in 2015–2019. Only legionellosis was more frequent throughout the year than that in 2015–2019. Lower activity was also observed in some contact transmitted, airborne-transmitted, and fecal-oral transmitted diseases. However, carbapenem-resistant Enterobacteriaceae, exanthema subitum, showed the same trend as the previous 5 years. In conclusion, our study has shown that public health interventions for the COVID-19 pandemic may have effectively prevented the transmission of most droplet-transmitted diseases and those transmitted through other routes.

• I don’t understand what the sentence on lines 68-69 is meant to indicate. Please rephrase or expand here.

Reply to reviewer: We revised the pointed out sentence as follows;

Datasets about COVID-19 (pointed out sentences of page 4, line 68-71)

Datasets on COVID-19 were extracted from the domestic infection status officially 68 released by the Ministry of Health, Labourand Welfare of Japan based on the NESID [7]. The COVID-19 patients were diagnosed using polymerase chain reaction (PCR) or antigen testing for SARS-CoV-2.

 ↓

Datasets about COVID-19 (revised)

The COVID-19 pandemic from January 16, 2020, to December 31, 2020 was demonstrated by datasets from the National Epidemiological Surveillance of Infectious Diseases (NESID) under the Infectious Diseases Control Law. We obtained the open data about COVID-19 from January 16, 2020 to December 31, 2020 from the Ministry of Health, Labour and Welfare [7]. Then, data on the daily number of new positive cases who have taken polymerase chain reaction (PCR) for SARS-CoV-2 or antigen testing for SARS-CoV2 were used in this study. These domestic cases do not include cases of airport quarantine.

• In the methods, more details are needed on why the inclusion/exclusion criteria were applied. Why was 400 chosen as the minimum number of cases? Why were fulminant, invasive, and enteric infections other than rotavirus excluded?

Reply to the reviewer: In this paper, we intend to cover and report on many infectious diseases. However, it is also true that the number of cases of infectious diseases reflects that it is an important disease. Since there are 365 days in a year, we set the number to more than 400, considering more than one case per day. It is also true that the total number of invasive infections is small, but it is interpreted that invasive infections indicate the severity of the disease and do not necessarily reflect the frequency or route of infection. Therefore, we deleted those from this analysis.

 Material and Methods (corrected the sentences on lines 73 to 78 from page 4 to page 5)

Common infectious diseases were selected from the nationally notifiable diseases according to the following: i) we were excluded the diseases with l<400 cases of infection per year. Since there are 365 days in a year, we set the number to >400, considering more than one case per day. ii) We excluded fulminant and invasive infectious diseases, such as invasive pneumonia disease, invasive meningococcal disease, and severe invasive streptococcal disease. The total number of invasive infections is small; however, invasive infections indicate the severity of the disease and do not necessarily reflect the frequency or route of infection. Therefore, we deleted those from the analysis. iii) We excluded “infectious gastroenteritis”, which is a syndrome that induced by various causes such as bacteria, viruses, parasites. Difficulties arise when classifying it via transmission route. Therefore excluded. iv) In addition, we excluded monthly reports of infections, such as gonococcal infections or multi-drug-resistant Pseudomonas aeruginosa infection.

• I suggest using “common infectious diseases” rather than “representative infectious diseases” throughout the manuscript. If “representative” is meant to indicate something other than “common”, please explain further.

Reply to reviewer: We revised “representative infectious diseases” throughout the manuscript as follows:

P2L27, P4L73, P4L58-59, P11L138, P12L149, P12L141, P14L195: 

representative common infectious disease

↓

common infectious disease

• I suggest using the phrase “fecal-oral transmission” instead of “oral transmission”.

Reply to the reviewer: We revised according to your suggestion as follows

L98, 102, 210, 377, Table 2, Fig.3

oral transmission

↓

fecal-oral transmission

• In Figure 1, I suggest using the y-axis label “Number of cases per day”.

Reply to the reviewer: We revised according to your suggestion as follows:

　　　　

The label of y-axis in Figure 1: Number of patients per day

↓

The label of y-axis in Figure 1: Number of cases per day 

• In Figures 2-3, please use colorblind-friendly colors instead of green and red.

Reply to the reviewer: Thank you very much for your comment. We used gray, red and blue color with light gray background color in Fig 2. Similarly, we used a combination of blue and red in Fig 3.

My revision

We needed support for statistical analysis in this revision. Therefore, I added two authors for this paper.

Hiroyoshi Iwata, Clinical Pharmacology & Therapeutics, University of the Ryukyus School of Medicine 

Shinichiro Ueda, Clinical Pharmacology & Therapeutics, University of the Ryukyus School of Medicine

We changed “Varicella” to “Chicken pox” because the name of “Chicken pox” is more common. Similarly, pertussis is sometimes used as the name of “Whooping cough”, so it is shown in parentheses where necessary. 

In this revision, we added supplemental table and figure. Therefore, I described the supporting information.

Supporting information

S1 Fig. Cross-correlation graph. S1A: Cross-correlation between COVID-19, and common　infectious diseases under the national sentinel surveillance. S1B: Cross-correlation between COVID-19, and common infectious diseases under the national notifiable disease surveillance. The X-axis indicates the correlation coefficient between COVID-19 and each common infectious disease. The Y-axis indicates Lag (week).

S1 Table. Case numbers per week about COVID-19 and case number per week per sentinel cite of notifiable infectious diseases. These weekly numbers of COVID-19 were calculated from the daily number of the new positive cases that have taken polymerase chain reaction for SARS-CoV-2 or antigen testing for SARS-CoV2. The data was obtained from the Ministry of Health, Labour and Welfare [7]. The numbers of notifiable infectious diseases were collected from the Infectious Diseases Weekly Reports in accordance with the National Epidemiological Surveillance of Infectious disease [8].

S2 Table. The total number of out-of-hospital (outpatient and home medical care) cases in the medical clinics and insurance pharmacies compared with that in the same month of the previous year. The number of cases referred to here is the number of medical fee statements (receipts), and each medical institution prepares one statement for one patient every month. The data is national data and are obtained from the Ministry of Health, Labor and Welfare's estimated medical expenses database [11].

---

## [Decision Letter · Decision Letter 1]

8 Oct 2021

PONE-D-21-11519R1Activity of common infectious diseases in Japan during the COVID-19 pandemicPLOS ONE

Dear Dr. Hibiya,

Thank you for submitting your manuscript to PLOS ONE. After careful consideration, we feel that it has merit but does not fully meet PLOS ONE’s publication criteria as it currently stands. Therefore, we invite you to submit a revised version of the manuscript that addresses the points raised during the review process.

We look forward to receiving your revised manuscript.

Kind regards,

Martial L Ndeffo Mbah, Ph.D

Academic Editor

PLOS ONE

Journal Requirements:

Reviewers' comments:

Reviewer's Responses to Questions

**Comments to the Author**

1. If the authors have adequately addressed your comments raised in a previous round of review and you feel that this manuscript is now acceptable for publication, you may indicate that here to bypass the “Comments to the Author” section, enter your conflict of interest statement in the “Confidential to Editor” section, and submit your "Accept" recommendation.

Reviewer #1: All comments have been addressed

Reviewer #2: (No Response)

2. Is the manuscript technically sound, and do the data support the conclusions?

Reviewer #1: Yes

Reviewer #2: Yes

3. Has the statistical analysis been performed appropriately and rigorously? 

Reviewer #1: Yes

Reviewer #2: Yes

4. Have the authors made all data underlying the findings in their manuscript fully available?

Reviewer #1: Yes

Reviewer #2: Yes

5. Is the manuscript presented in an intelligible fashion and written in standard English?

Reviewer #1: Yes

Reviewer #2: Yes

6. Review Comments to the Author

Reviewer #1: In general, the revision adequately addresses the issues I had raised in my initial review. I have only minor comments, that might further be addressed:

1. Incidence rather than activity, in the title?

2. Throughout the text, lag rather than rag, when referring to cross-correlations.

3. The authors might mention the impact of mumps vaccine. I suspect the authors’ comments concerning measles vaccine would also bear on mumps.

4. A methodological issue: the authors might mention in the methods what range of values was used when calculating moving averages (e.g., 5 point moving average?)

5. One upshot of cross-correlation values is “interpretation” of the lag (or, range of lags) that leads to maximal or minimal cross-correlation values. E.g., one might speculate that periods of high COVID-19 incidence might in some sense tamp down or delay reported incidence of certain other diseases. Which diseases are or are not so affected would also be of interest.

6. The difference in differences methodology makes some strong assumptions on the putative relationship between the two series. The authors might comment on the validity of these assumptions in their context.

Reviewer #2: The authors have very thoroughly addressed my comments. The difference-in-difference analysis helps to show a clear story of reduced reporting of infections transmitted by droplets in Japan in 2020. The analysis and discussion of changes in outpatient behavior is also very helpful for understanding the context of broader health seeking during the pandemic. I only have a few remaining minor points:

1) The cross-correlation results in the abstract lines 36 to 38 and results lines 256 to 257 are hard to follow. What does rag -1 week mean? I suggest using more common language there. Please also provide a definition for what the numbers provided in parentheses represent, and limit to 2 decimal places. In addition, what does “among them” on line 257 refer to if there were no correlations with COVID-19 as stated in the previous sentence (or perhaps that is a typo?). It would also be helpful to include a short explanation of what the implications/importance of this result is, and/or what the rationale was for performing this analysis.

2) I suggest removing the sentence in lines 311 to 312, unless the authors can identify the reason that the CCF analysis “didn’t work”.

3) There are several instances in the text where it sounds like the authors are implying a causal link between COVID-19 measures and reductions in reported cases. For example lines 435 to 436: “Therefore, the COVID-19 control measures did not affect the spread of legionellosis.” The analysis in this study is descriptive, not causal, so this section could read something like: “COVID-19 control measures were not associated with changes in reported cases of legionellosis, which is consistent with the transmission route.. etc.”. Other examples where this should be clarified include lines 426-427, 442-444, 500-501, 511-513. In addition, the titles of these sections (currently: “Effect of COVID-19 measures on the incidence...”) should be revised to read something like: “Associations between COVID-19 measures and incidence of XX” or “Patterns in incidence of XX pathogens during the 2020 COVID-19 pandemic”.

7. PLOS authors have the option to publish the peer review history of their article (what does this mean?). If published, this will include your full peer review and any attached files.

Reviewer #1: **Yes: **James Koziol

Reviewer #2: No

---

## [Author Response · Author response to Decision Letter 1]

13 Nov 2021

6. Review Comments to the Author

Reviewer #1: In general, the revision adequately addresses the issues I had raised in my initial review. I have only minor comments that might further be addressed:

1. Incidence rather than activity, in the title?

Reply to reviewer: I changed the word "activity" to "incidence" in the title according to the reviewer's suggestion as follows: 

Title page (Corrected the first line of text on page 1) 

Incidence of common infectious diseases in Japan during the COVID-19 pandemic

2. Throughout the text, lag rather than rag, when referring to cross-correlations.

Reply to reviewer: I duly changed the word "rag" to "lag" according to the reviewer’s comments. 

3. The authors might mention the impact of mumps vaccine. I suspect the authors’ comments concerning measles vaccine would also bear on mumps.

Reply to reviewer: As the reviewer pointed out, it is true that there are populations that are susceptible to measles, rubella, and mumps. Therefore, the same discussion can be made for mumps and rubella. We duly revised the discussion about rubella and mumps as follows:

Discussion (revised on Page 25, Line 414 to Page 26, Line 431)

In the present study, rubella and HFMD showed a marked decrease throughout the year compared with 2015–2019. There may be several possible explanations for this. For example, in 2013, reported rubella cases were 14,344. However, a decline was noted in subsequent years, with 319 cases in 2014, 163 in 2015, 126 in 2016, and 93 in 2017 [30]. Since August 2018, there has been a rapid increase in rubella incidence: 2,941 in 2018 and 2,306 in 2019 [30]. The total number of rubella cases in 2020 was 100. This means that the epidemic may have been contained, and it is thought to have just returned to baseline, without the influence of COVID-19. In this study, the CCF of rubella showed a negative correlation coefficient over lag 0–lag 20 weeks, suggesting that it might be suppressed over a long period. Incidentally, vaccination is effective in preventing rubella. The Meals-Mumps-rubella (MMR) vaccine has been introduced in many countries worldwide. However, in Japan, the MMR vaccine has not been introduced due to side effects. In addition, there are generations with low vaccination rates for rubella (males born between 1962 and 1978). The rubella epidemic of 2018-19 mainly affected men in the indicated generations [30]. This suggests that Japan is still at risk for rubella epidemics in these susceptible populations. Therefore, the lower activity throughout the year may have been influenced by preventative behavioral changes associated with the COVID-19 epidemic. 

 Discussion (revised on Page 26, Line 439 to Line 449)

 The present study showed that the mumps epidemic curve was stable. The patient count of mumps has been below 0.2 per sentinel site since 2018 and has been on a downward trend each year [8]. According to the present CCF analysis of mumps, the strongest negative correlation was obtained at lag 1 week. Vaccination against mumps has also become compulsory in many countries. However, the vaccination is still voluntary in Japan. According to the National Epidemiological Surveillance of Vaccine-Preventable Diseases, the vaccination coverage in recent years was 30~ 40% and the antibody coverage using sera stored in domestic serum banks was approximately 70% [32]. This means that there is a population that is susceptible to mumps in Japan. Therefore, the epidemic may have been controlled by the behavioral change in this susceptible population. 

32. National Institute of Infectious Diseases. Mumps (infectious parotitis) in Japan, as of September 2016. IASR 2016;37: 185-186. Available from: 

https://www.niid.go.jp/niid/images/idsc/iasr/37/440e.pdf

4. A methodological issue: the authors might mention in the methods what range of values was used when calculating moving averages (e.g., 5 point moving average?)

Reply to reviewer: The moving averages were calculated with 2 points. Therefore, we mentioned it in the materials and methods as follows;

. 

 Material and Methods (inserted on Page 12, Line 146 to Line 147)

For the epidemic curves of "COVID-19" and "common infectious diseases:, a moving average with 2 points was calculated. 

5. One upshot of cross-correlation values is “interpretation” of the lag (or, range of lags) that leads to maximal or minimal cross-correlation values. E.g., one might speculate that periods of high COVID-19 incidence might in some sense tamp down or delay reported incidence of certain other diseases. Which diseases are or are not so affected would also be of interest.

Reply to reviewer: We have duly added an interpretation of the lag (or range of lags) in the materials and methods and revised the results and discussion of the cross-correlation function as follows:

 Material and Methods (revised on Page 12, Line 147 to Line 154)

Statistical analysis

The cross-correlation function (CCF) was used to understand whether there was a time-lagged correlation between common infectious disease count and COVID-19 incidence. CCF is a function that expresses the similarity between two-time series and gives information about how similar and displaced one-time series is to the other. CCF takes a value in the range of -1 (negative correlation) to 1 (positive correlation). If the correlation value exceeds the confidence level, then the two series are correlated. The cross-correlation between the two variables is statistically significant at approximately the 5% level of significance. 

Result (Page 17, Line 253 to Page 18, Line 265)

Additionally, we examined the CCF between the two-time series of COVID-19 and common infectious disease counts. The incidence of scrub typhus peaked significantly 1 week earlier than the third peak of COVID-19 incidence (correlated efficient=0.87). Herpangina showed a significant peak 15 weeks earlier than the second peak time of COVID-19 incidence (cross-correlation values=0.65). The strongest negative correlation (cross-correlation values=0.40) was obtained at lag 1 week for mumps. Influenza, group A Streptococcal pharyngitis, erythema infectiosum, epidemic keratoconjunctivitis, Mycoplasma pneumoniae pneumonia, infectious gastroenteritis (rotavirus), pertussis, rubella, measles showed a negative correlation with COVID-19 in the lag from minus 20 weeks to 0 weeks. Sexually transmitted diseases, including amoebiasis, AIDS, and syphilis, reached their lowest peaks 1 to 2 weeks later than the peak in COVID-19 incidence. A closer look at the epidemic curve showed a phenomenon with a slight deviation from each peak of COVID-19. 

 Discussion (inserted on Page 27, Line 460 to Line 463)

Keratoconjunctivitis showed a negative correlation against COVID-19 in the lag from minus 20 weeks to 0 weeks. This may suggest that the hand-washing and hand-disinfection practices triggered by the COVID-19 epidemic also had a long-term effect on reducing the prevalence of contact infections.

Discussion (inserted on Page 33, Line 594 to Page 34, Line 599)

In this study, we also examined the CCFs between the two-time series of scrub typhus and COVID-19 disease counts. The peak incidence of COVID-19 coincided with the peak incidence of scrub typhus exhibiting a 1-week lag. Although there is a large seasonal bias in the occurrence of scrub typhus disease, we cannot deny the possibility that the release from behavioral inhibition by COVID-19 epidemic led to an increase in the incidence of scrub typhus infection.

Discussion (inserted on Page 35, Line 635 to Page 36, Line 638)

The lowest peak of current incidents of sexual transmitted disease showed a lag of few weeks than the peak of COVID-19 by cross-correlation analysis. The results may support the possibility that people became fearful when they saw the rapid increase in the number of new COVID-19 cases and changed their behavior.

6. The difference in differences methodology makes some strong assumptions on the putative relationship between the two series. The authors might comment on the validity of these assumptions in their context.

Reply to reviewer: We duly added the comment on the validity of these assumptions in the text.

In the section of Material and Methods (inserted on Page 13, Line 161 to Line 166)

We made two assumptions for the difference-in-difference linear regression in our preliminary experiments. First, the parallel trend assumption was valid for both incidences because the current incidents and incidences of the previous 5 years of common infections were parallel. The common shock assumption was also valid, as it showed a similar change when an event (the epidemic of COVID-19) occurred, indicated the appropriateness of this study design. 

Reviewer #2: The authors have very thoroughly addressed my comments. The difference-in-difference analysis helps to show a clear story of reduced reporting of infections transmitted by droplets in Japan in 2020. The analysis and discussion of changes in outpatient behavior is also very helpful for understanding the context of broader health seeking during the pandemic. I only have a few remaining minor points:

1) The cross-correlation results in the abstract lines 36 to 38 and results lines 256 to 257 are hard to follow. What does rag -1 week mean? I suggest using more common language there. Please also provide a definition for what the numbers provided in parentheses represent, and limit to 2 decimal places. In addition, what does “among them” on line 257 refer to if there were no correlations with COVID-19 as stated in the previous sentence (or perhaps that is a typo?). It would also be helpful to include a short explanation of what the implications/importance of this result is, and/or what the rationale was for performing this analysis.

Reply to reviewer: Apologies for mistakenly using the word "rag" for "lag". We duly revised the results of cross-correlation in the abstract lines 36 to 38 and the results lines 256 to 257. In the Summary and the Results, "CCF" has been reworded to cross-correlation values, and numbers in parentheses have been limited to two decimal places. We have also reviewed the overall description of cross-correlation functions and have then included an explanation of “lag” and the benefits that can be derived from cross-correlation functions in the Material and Methods.

Abstract (revised on Page 2, Line 33 to Line 36)

The cross-correlation functions between the two-time series of common infectious and COVID-19 disease counts were examined. Many droplet-borne diseases, infectious gastroenteritis (rotavirus), and measles showed a negative correlation against COVID-19 in the lag from minus 20 weeks to 0 weeks. 

Material and Methods (inserted on Page 12, Line 147 to Line 154)

 Statistical Analyses

The cross-correlation function (CCF) was used to understand whether there was a time-lagged correlation between common infectious disease count and COVID-19 incidence. CCF is a function that expresses the similarity between two time series, and gives information about how similar and displaced one time series is to the other. CCF takes a value in the range of -1 (negative correlation) to 1 (positive correlation). If the correlation value exceeds the confidence level, then the two series are correlated. The cross-correlation between the two variables is statistically significant at approximately 5% level of significance. 

Result (revised on Page 17, Line 253 to Page 18, Line 265)

Additionally, we examined the CCF between the two-time series of COVID-19 and common infectious disease counts. The incidence of scrub typhus peaked significantly 1 week earlier than the third peak of COVID-19 incidence (cross-correlation values=0.87). Herpangina showed a significant peak 15 weeks earlier than the second peak time of COVID-19 incidence (cross-correlation values=0.65). The strongest negative correlation (cross-correlation values=0.40) was obtained at lag 1 week for the mumps. Influenza, group A Streptococcal pharyngitis, erythema infectiosum, epidemic keratoconjunctivitis, Mycoplasma pneumoniae pneumonia, infectious gastroenteritis (rotavirus), pertussis, rubella, measles showed a negative correlation against COVID-19 in the lag from minus 20 weeks to 0 weeks. Sexually transmitted diseases, including amoebiasis, AIDS, and syphilis, reached their lowest peaks 1 to 2 weeks later than the peak in COVID-19 incidence. A closer look at the epidemic curve showed a phenomenaon with a slight deviation from each peak of COVID-19.

2) I suggest removing the sentence in lines 311 to 312, unless the authors can identify the reason that the CCF analysis “didn’t work”.

Reply to reviewer: We duly deleted the sentence in page 19, lines 311 to 312 in the second submitted manuscript.

3) There are several instances in the text where it sounds like the authors are implying a causal link between COVID-19 measures and reductions in reported cases. For example lines 435 to 436: “Therefore, the COVID-19 control measures did not affect the spread of legionellosis.” The analysis in this study is descriptive, not causal, so this section could read something like: “COVID-19 control measures were not associated with changes in reported cases of legionellosis, which is consistent with the transmission route. etc.”. Other examples where this should be clarified include lines 426-427, 442-444, 500-501, 511-513. In addition, the titles of these sections (currently: “Effect of COVID-19 measures on the incidence...”) should be revised to read something like: “Associations between COVID-19 measures and incidence of XX” or “Patterns in incidence of XX pathogens during the 2020 COVID-19 pandemic”.

Reply to reviewer: In accordance with the reviewer's suggestion, we have removed all sentences that implied a causal relationship. The following text has been removed. 

Discussion (Second submitted manuscript: Page 25, Line435-436)

Therefore, the COVID-19 control measures did not affect the spread of legionellosis.

Discussion (Second submitted manuscript: Page 25, Line426-427)

In the present study, we considered that preventive measures for COVID-19 suppressed the small epidemic of HFMD caused by coxsackievirus A16

Discussion (Second submitted manuscript: Page 26, Line442-444)

Therefore, we considered that behavior modification, such as hand hygiene, for preventing COVID-19 was also effective against epidemic keratoconjunctivitis.

Discussion (Second submitted manuscript: Page 28, Line 500-501)

Nevertheless, the suppression of the measles epidemic without any outbreaks demonstrates the effectiveness of public health interventions.

Discussion (Second submitted manuscript: Page 29, Line 511-513)

Nevertheless, the present results suggest that infectious gastroenteritis caused by rotavirus can be prevented by proper hand washing, disinfection, gargling, washing, and sterilization in household settings. 

Reply to reviewer: In response to the reviewers' comments, we revised the title of each section have duly made the following amendments to each section. 

The title of each section

Page22, Line 364 - Line 365: Associations between COVID-19 measures and the incidence of droplet-transmitted diseases

Page27, Line 455 - 456: Associations between COVID-19 measures and the incidence of contact-transmitted diseases

Page 28, Line 478 - 479: Associations between COVID-19 measures and the incidence of airborne transmitted diseases

Page 30, Line 525 - 526: Associations between COVID-19 measures and the incidence of fecal orally transmitted diseases

Page 33, Line 585 - 586: Associations between COVID-19 measures and the incidence of vector-borne diseases

Page 34, Line 601 -602: Associations between COVID-19 measures and the incidence of STDs

Author’s corrections: In line with this revision, the authors have duly made the following corrections to the discussion of measles/rotavirus infections.

Discussion (inserted on Page 30, Line 515 to Line 523)

The measles vaccine has been said to provide lifelong immunity; however, it is known that antibody levels decrease even after vaccination in the absence of recent epidemics [46]. For this reason, even in Japan, there are populations with low antibody levels, and small outbreaks can occur at any time. The overall measles antibody prevalence in 2020 was 96.3%, although the antibody prevalence among children aged 1 year in 2020 was 69.8%, a significant decrease from 81.6% in 2019 [47]. This has been attributed to a temporary abstaining from vaccination [47]. However, the impact on the measles epidemic is thought to have been short-term and small. In addition, the estimated infected areas of the 12 measles cases reported in 2020 were 7 national, 4 overseas, and 1 unknown [47]. While the number of domestic cases has decreased, the proportion of imported cases is becoming more apparent. If there had been no entry restrictions due to the COVID-19 pandemic, such imported cases might have increased.

47) National Institute of Infectious Disease. Measles in Japan, as of June 2021. IASR. 42:177-182, 2021. Available from: https://www.niid.go.jp/niid/en/typhi-m/iasr-reference/1687-iasr-backnomber-e.html

Discussion (inserted on Page 31, Line 533 to Line 537)

Among fecal-oral infections, only rotavirus infection showed a negative correlation between lag 0 and 25 weeks in the CCFs analysis. This suggests that rotavirus infection generally occurs in the home rather than in eating establishments and that appropriate hygiene behavior in the home may have had a long-term effect.

---

## [Decision Letter · Decision Letter 2]

1 Dec 2021

Incidence of common infectious diseases in Japan during the COVID-19 pandemic

PONE-D-21-11519R2

Dear Dr. Hibiya,

We’re pleased to inform you that your manuscript has been judged scientifically suitable for publication and will be formally accepted for publication once it meets all outstanding technical requirements.

Kind regards,

Martial L Ndeffo Mbah, Ph.D

Academic Editor

PLOS ONE

Additional Editor Comments (optional):

Reviewers' comments:

Reviewer's Responses to Questions

**Comments to the Author**

1. If the authors have adequately addressed your comments raised in a previous round of review and you feel that this manuscript is now acceptable for publication, you may indicate that here to bypass the “Comments to the Author” section, enter your conflict of interest statement in the “Confidential to Editor” section, and submit your "Accept" recommendation.

Reviewer #1: All comments have been addressed

Reviewer #2: All comments have been addressed

2. Is the manuscript technically sound, and do the data support the conclusions?

Reviewer #1: Yes

Reviewer #2: Yes

3. Has the statistical analysis been performed appropriately and rigorously? 

Reviewer #1: Yes

Reviewer #2: Yes

4. Have the authors made all data underlying the findings in their manuscript fully available?

Reviewer #1: Yes

Reviewer #2: Yes

5. Is the manuscript presented in an intelligible fashion and written in standard English?

Reviewer #1: Yes

Reviewer #2: Yes

6. Review Comments to the Author

Reviewer #1: (No Response)

Reviewer #2: My comments pertaining to analysis/results/discussion have been addressed. I have a final editorial/clarification suggestion:

In the abstract, the section describing the cross-correlation is still too technical and does not describe what the findings of this analysis actually mean: "The cross-correlation functions between the two-time series of common infectious and COVID-19 disease counts were examined. Many droplet-borne diseases, infectious gastroenteritis (rotavirus), and measles showed a negative correlation against COVID-19 in the lag from minus 20 weeks to 0 weeks."

The "lag minus 20 weeks" portion can be included in the methods/results, but the abstract should read something along the lines of: "We examined correlations over time using a cross-correlation analysis. We found that weekly cases of measles, rotavirus, and many droplet-borne diseases were negatively correlated with with COVID-19 cases up to 20 weeks in the past".

7. PLOS authors have the option to publish the peer review history of their article (what does this mean?). If published, this will include your full peer review and any attached files.

Reviewer #1: **Yes: **James Koziol

Reviewer #2: No

---

## [Editor Report · Acceptance letter]

3 Jan 2022

PONE-D-21-11519R2 

Incidence of common infectious diseases in Japan during the COVID-19 pandemic 

Dear Dr. Hibiya:

I'm pleased to inform you that your manuscript has been deemed suitable for publication in PLOS ONE. Congratulations! Your manuscript is now with our production department. 

Kind regards, 

on behalf of

Dr. Martial L Ndeffo Mbah 

Academic Editor

PLOS ONE